# Direct and indirect Z-scheme heterostructure-coupled photosystem enabling cooperation of CO$_2$ reduction and H$_2$O oxidation

Ying Wang [1,2], Xiaotong Shang[1], Jinni Shen[1], Zizhong Zhang [1✉], Debao Wang[2], Jinjin Lin[1], Jeffrey C. S. Wu [3✉], Xianzhi Fu[1], Xuxu Wang [1✉] & Can Li [4✉]

The stoichiometric photocatalytic reaction of CO$_2$ with H$_2$O is one of the great challenges in photocatalysis. Here, we construct a Cu$_2$O-Pt/SiC/IrO$_x$ composite by a controlled photo-deposition and then an artificial photosynthetic system with Nafion membrane as diaphragm separating reduction and oxidation half-reactions. The artificial system exhibits excellent photocatalytic performance for CO$_2$ reduction to HCOOH and H$_2$O oxidation to O$_2$ under visible light irradiation. The yields of HCOOH and O$_2$ meet almost stoichiometric ratio and are as high as 896.7 and 440.7 µmol g$^{-1}$ h$^{-1}$, respectively. The high efficiencies of CO$_2$ reduction and H$_2$O oxidation in the artificial system are attributed to both the direct Z-scheme electronic structure of Cu$_2$O-Pt/SiC/IrO$_x$ and the indirect Z-scheme spatially separated reduction and oxidation units, which greatly prolong lifetime of photogenerated electrons and holes and prevent the backward reaction of products. This work provides an effective and feasible strategy to increase the efficiency of artificial photosynthesis.

[1] State Key Laboratory of Photocatalysis on Energy and Environment, Research Institute of Photocatalysis, College of Chemistry, Fuzhou University, 350108 Fuzhou, China. [2] Key Lab of Inorganic Synthetic and Applied Chemistry, State Key Lab Base of Eco-Chemical Engineering, College of Chemistry and Molecular Engineering, Qingdao University of Science & Technology, 266042 Qingdao, China. [3] Department of Chemical Engineering, National Taiwan University, 10617 Taipei, Taiwan. [4] State Key Laboratory of Catalysis, Dalian Institute of Chemical Physics, Chinese Academy of Sciences, 116023 Dalian, China. ✉email: z.zhang@fzu.edu.cn; cswu@ntu.edu.tw; xwang@fzu.edu.cn; canli@dicp.ac.cn

Solar-driven photocatalytic conversion of carbon dioxide ($CO_2$) to valuable organics or solar fuel is an attractive solution to both current energy and environment problems[1–4]. Reduction of $CO_2$ under visible light accounting for 45% sunlight energy by water rather than organic compound as an electron donor is the ultimate goal of photocatalysis. Enormous efforts have been devoted to developing a highly efficient photocatalyst for this prospect[5–7]. However, so far none of photocatalyst systems is satisfactory. Development of novel photocatalyst or system to realize highly efficient conversion of $CO_2$ remains the focus of future research.

In various reduction products of $CO_2$, including CO, HCOOH, $CH_3OH$, HCHO, $CH_4$, etc., HCOOH is a chemical with wide applications[8], and even is considered as a promising biorenewable feedstock for fine chemicals[9]. Although HCOOH as a two-electron-transfer product has the lowest degree of reduction among conversion products of $CO_2$, all the reported photocatalysts to date still suffer from very low efficiency for HCOOH similar to other organic products[10–13]. This implies that the photocatalytic reduction of $CO_2$ to HCOOH is not easier than the reduction to other products. It is well known that the photocatalytic conversion of $CO_2$ with $H_2O$ involves two half-reactions, i.e. the reduction of $CO_2$ by the photogenerated electrons and protons, and the oxidation of $H_2O$ by the photogenerated holes. However, most of the reported photocatalysts could not catalyse simultaneously reduction of $CO_2$ to HCOOH and oxidation of $H_2O$ to $O_2$[14], and only work in the presence of organic hole-scavengers (e.g. triethanolamine (TEOA), trimethylamine (TEA), or ethylenediaminetetraacetic acid (EDTA))[15,16]. Such photocatalytic $CO_2$ reduction at the cost of sacrificial electron donors is not sustainable and likely economically unsound. Even using hole scavenger, the photocatalysts only exhibit a formation rate of HCOOH with tens of micromoles, typically such as some metal-organic framework (MOF) materials ($NH_2$-MIL-125(Ti), MIL-101(Fe))[17,18], inorganic–organic hybrid materials (a binuclear ruthenium(II) complex coupled with $Ag/C_3N_4$ (ref. [19,20]), Cu(I) complex photosensitized Mn(I) complex catalysts[21]), and metal sulfide semiconductors $((Mo−Bi)S_x/CdS)$[22]. The C and Fe co-doped $LaCoO_3$ was reported to display an HCOOH yield up to 128 $\mu mol\,g^{-1}\,h^{-1}$ without sacrificial reagent, but the oxidation product $O_2$ was not analysed[23]. Such photocatalytic $CO_2$ reduction without accompanying oxidation half-reaction is inexplicable. The photosynthesis essentially requires the stoichiometric photocatalytic $CO_2$ reduction and $H_2O$ oxidation which remains a great challenge in photocatalysis[24].

Silicon carbide (SiC), a metal-free semiconductor material, possesses a moderate wide band gap (2.4 eV) with an enough negative CB (ca. −1.1 V) to satisfy multielectron reactions of $CO_2$ reduction with $H_2O$ into carbon fuel and oxygen by solar energy[25,26]. So it has been considered as a promising photocatalyst for $CO_2$ conversion since the early research work[27]. However, the expected photocatalytic efficiency has not been achieved so far. This is due to very large difference between electron and hole migration rates in SiC (electron mobility 700 $cm^2\,V^{-1}\,s^{-1}$, hole mobility 90 $cm^2\,V^{-1}\,s^{-1}$), which leads to the accumulation of photogenerated holes in the bulk and in turn suppresses the further generation of electrons under light irradiation. This makes the photogenerated carriers to be short-lived, especially the oxidation ability to be poor[28,29]. Moreover, the pristine SiC is lack of active sites for $CO_2$ adsorption and activation. Therefore, it is desirable to find the suitable cocatalysts to modify SiC. Additionally, the thermodynamically favourable backward reaction of the produced organics with oxygen on the photocatalyst surface is detrimental to decrease efficiency of $CO_2$ with pure $H_2O$ in the conversional one-pot reaction. These problems can be concurrently solved by constructing the multi-photocatalyst integration systems in which the oxidation and reduction reactions are independent in space but coupled in the transfer of photogenerated charges.

Here, we report a $Cu_2O–Pt/SiC/IrO_x$ hybrid photocatalyst, which is prepared by loading the photo-oxidation unit ($IrO_x$) and the photoreduction unit ($Cu_2O–Pt$) on SiC surface. This configuration can enhance the lifetime of photogenerated charges and the $CO_2$ adsorption, and thus the photocatalytic efficiency. Furthermore, we construct a spatially separated reaction system consisting of two reaction chambers analogous to the natural photosynthetic systems. One chamber is loaded with the $Cu_2O–Pt/SiC/IrO_x$ photocatalyst and $Fe^{2+}$ for $CO_2$ reduction, while the other chamber with the known $Pt/WO_3$ and $Fe^{3+}$ for $H_2O$ oxidation, and the two chambers are divided by a Nafion membrane that allows $Fe^{2+}$ and $Fe^{3+}$ ions to permeate through. This design facilitates $H_2O$ oxidation half-reaction and suppresses the backward reaction of the products. For the photocatalytic reaction of $CO_2$ with $H_2O$ to HCOOH and $O_2$, the system shows very high photocatalytic efficiency under visible light irradiation. The HCOOH yield is as high as 896.7 $\mu mol\,g^{-1}\,h^{-1}$ for the long-term reaction, 527 times higher than that of the pristine SiC (1.7 $\mu mol\,g^{-1}\,h^{-1}$) in the conversional one-pot reaction. Most importantly, $O_2$ with a stoichiometric ratio is evolved concurrently. To the best of our knowledge, such high activity for reaction of $CO_2$ with pure $H_2O$ to HCOOH and $O_2$ is rarely reported before.

## Results and discussion

**Configuration and composition of photocatalysts.** Figure 1 illustrates the formation process of the photocatalyst $Cu_2O–Pt/SiC/IrO_x$ through the step-by-step photodeposition of Pt, $Cu_2O$ and $IrO_x$ on 3C-SiC (face centre cubic phase of SiC) surface. First, Pt nanoparticles were loaded onto SiC by a simple photodeposition to obtain Pt/SiC sample. Then the resulting Pt/SiC samples were dispersed in aqueous solution containing both $Cu^{2+}$ and $IrCl_6^{3−}$ ions with UV light illumination, which led $Cu^{2+}$ to reduction into $Cu_2O$ species and $IrCl_6^{3−}$ to oxidation into $IrO_x$. Due to higher work function (5.6 eV) of Pt, the photogenerated electrons on SiC migrated to Pt and the photogenerated holes remained on SiC[30,31]. Thus the reduction reaction occurred on the Pt particle, while the oxidation reaction did on SiC. We thus concluded that the $Cu_2O$ was deposited on the Pt, while the $IrO_x$ on SiC. The resulting $Cu_2O–Pt$ and $IrO_x$ were located at different region of SiC surface. For comparison, the two reference samples, $Cu_2O–Pt/SiC$ and $Pt/SiC/IrO_x$, were prepared also in the similar conditions with $Cu^{2+}$-contained solution and $IrCl_6^{3−}$-contained solution, respectively. The loading amounts of cocatalysts on SiC samples were controlled by the photodeposition time (0.5–15 h) and then were quantified by a quadrupole inductively coupled plasma mass spectrometry (ICP-MS), as summarized in Supplementary Table 1 and Supplementary Note 1. The contents of Pt, $Cu_2O$ and $IrO_x$ on the samples were controlled in the range of 0.83–2.6, 0.52–2.7 and 0.87–3.2 wt%, respectively. Based on the photocatalytic $CO_2$ reduction results, the optimal amount of Pt, $Cu_2O$ and $IrO_x$ is ascertained to be 1.3%, 1.8% and 2.2 wt% for $Cu_2O–Pt/SiC/IrO_x$ photocatalyst, respectively. For the sake of brevity, hereafter the optimal photocatalyst with 1.3 wt% Pt, 1.8 wt% $Cu_2O$ and 2.2 wt% $IrO_x$ is

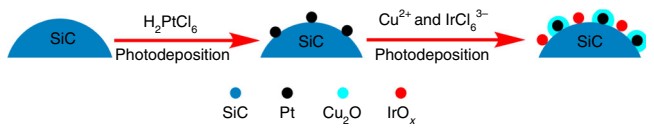

**Fig. 1 Formation process of $Cu_2O–Pt/SiC/IrO_x$ by photodeposition.** Schematic representation of $Cu_2O–Pt/SiC/IrO_x$ synthesis via controlled photodeposition.

referred simply to $Cu_2O$–Pt/SiC/$IrO_x$, unless specifically noted otherwise.

The samples were firstly characterized by X-ray powder diffraction (XRD) (Supplementary Fig. 1, Supplementary Note 2). For all samples, except the highly crystalline cubic phase SiC (JCPDS No. 65-0360), no XRD peaks corresponding to the cocatalysts (Pt, $Cu_2O$ and $IrO_x$) are observed due to their low contents and high dispersion on SiC surface. The BET specific surface area of SiC has a slight reduction from ~18 $m^2 g^{-1}$ for bare SiC to ~14 $m^2 g^{-1}$ for $Cu_2O$–Pt/SiC/$IrO_x$ (Supplementary Fig. 2, Supplementary Note 3), possibly because the cocatalysts with small-size particles block the micropore structure of SiC. The spatial locations of Pt, $Cu_2O$ and $IrO_x$ species on SiC surface were visualized by transmission electron microscopy (TEM) and HRTEM images (Fig. 2). The Pt nanoparticles have a uniform size of ca 2–4 nm and evenly distribute on the surface for all samples, as marked with yellow dotted circle (Fig. 2a, Supplementary Fig. 3). $Cu_2O$ was deposited over Pt nanoparticles to form a $Cu_2O$–Pt intimate contact configuration for both $Cu_2O$–Pt/SiC (Supplementary Fig. 3c, Supplementary Note 4) and $Cu_2O$–Pt/SiC/$IrO_x$ samples (Fig. 2a) as marked with red circle, while the $IrO_x$ species are not deposited at the same location as $Cu_2O$–Pt species for both Pt/SiC/$IrO_x$ (Supplementary Fig. 3d) and $Cu_2O$–Pt/SiC/$IrO_x$ samples (Fig. 2a), as marked with blue dotted circle. The lattice spacings of $Cu_2O$–Pt/SiC/$IrO_x$ samples (Fig. 2b, Supplementary Fig. 3) are 0.252, 0.226, 0.211 and 0.260 nm assigning to the (111) facet of SiC, (111) facet of Pt, (200) facet of $Cu_2O$ and (101) facet of $IrO_2$ (refs. [32–35]), respectively. The $Cu_2O$–Pt intimate contact structure is further testified by STEM-EDS mapping (Fig. 2c, Supplementary Fig. 4). In the selected area, Pt has the same distribution and appears at almost the same position as Cu, further demonstrating the deposition of $Cu_2O$ over Pt particles. Nevertheless, there is also a part of Cu to be deposited on SiC surface. $IrO_x$ looks like a random deposition on the entire surface of SiC due to very small cluster particles, but it

is separated from $Cu_2O$–Pt on the SiC surface, because the distribution of Ir-L in mapping images (Supplementary Fig. 5, Supplementary Note 4, Fig. 2c) has an obvious difference from that of other elements through careful comparison.

The chemical composition distribution of the outermost layer on SiC surface was further analysed by a high-sensitivity low-energy ion scattering (HS-LEIS) studies. 3 keV $^4He^+$ HS-LEIS spectra (Fig. 3a) give the signal of the light elements on the outer surface (such as C, O and Si), but have poor sensitivity to Cu, Pt and Ir heavy elements. Figure 3b shows the 5 keV $^{20}Ne^+$ HS-LEIS spectra of SiC, Pt/SiC, $Cu_2O$–Pt/SiC, Pt/SiC/$IrO_x$ and $Cu_2O$–Pt/SiC/$IrO_x$. Pt element is detected on the outmost surface for Pt/SiC. Only Cu element is observed on $Cu_2O$–Pt/SiC, clearly indicating that $Cu_2O$ is deposited on the surface of Pt nanoparticles. Although HS-LEIS peaks of Pt and Ir cannot be resolved using 5 keV $^{20}Ne^+$ because their atomic mass is too close, it should be noted that intensity of the fused peaks of Pt and Ir at 3367 eV for Pt/SiC/$IrO_x$ is significantly stronger than that of sole Pt peak in Pt/SiC. This indicates that both $IrO_x$ and Pt coexist on the outermost surface of Pt/SiC/$IrO_x$. However, the peak at 3367 eV for $Cu_2O$–Pt/SiC/$IrO_x$ samples weakens significantly as compared with Pt/SiC/$IrO_x$. Since Pt is covered by $Cu_2O$ in $Cu_2O$–Pt/SiC/$IrO_x$, the low peak at 3367 eV of $Cu_2O$–Pt/SiC/$IrO_x$ can be only assigned to $IrO_x$ on the outermost surface. Therefore, the HS-LEIS results constitute another strong evidence that the $Cu_2O$–Pt and $IrO_x$ cocatalysts are spatially separated on SiC surface for $Cu_2O$–Pt/SiC/$IrO_x$.

The chemical states of the Pt, Cu and Ir elements in samples were analysed by XPS (Fig. 4). Three sets of Pt 4f XPS peaks can be assigned to metallic $Pt^0$ and partially oxidized $Pt^{2+}$ and $Pt^{4+}$ species[36]. The ratio of $Pt^0$ is calculated to be accounting for 70 ± 4% of the sum Pt species for the all Pt-contained samples (Supplementary Table 2). Cu species and Ir species are mainly in the state of $Cu_2O$ and mixed valence oxides ($IrO_x$), respectively[37,38]. Nevertheless, the binding energies (BE) of Pt, Cu or Ir have obvious differences among samples. For Pt/SiC, the BE of Pt $4f_{7/2}$ and Pt $4f_{5/2}$ are respectively 70.8 and 74.1 eV, which are lower than that of the pure metallic Pt (Pt $4f_{7/2}$ = 71.2 eV)[39]. This is because the electron transfer occurs from the SiC substrate to Pt particles with higher work function[40,41]. When $Cu_2O$ is subsequently deposited on Pt/SiC to form $Cu_2O$–Pt/SiC, the BE of Pt $4f_{7/2}$ shifts towards higher energy (71.1 eV), but is still slight lower than that of metallic Pt, indicating the electron transfer still from SiC to $Cu_2O$ under mediation of Pt nanoparticles. On the contrary, when $IrO_x$ is deposited onto Pt/SiC to form Pt/SiC/$IrO_x$, the BE of Pt 4f shifts to lower position. This indicates that $IrO_x$ deposition induces the electron transfer from $IrO_x$ to the SiC surface and thus enhances the electron transfer to Pt particles. In the case of the co-deposition of $IrO_x$ and $Cu_2O$ on Pt/SiC, the BE of Pt 4f in $Cu_2O$–Pt/SiC/$IrO_x$ is comparable with $Cu_2O$–Pt/SiC. This demonstrates that the electrons are finally transferred from both SiC and $IrO_x$ into $Cu_2O$. As a result, the BE of Cu 2p for $Cu_2O$–Pt/SiC/$IrO_x$ is also lower than that of $Cu_2O$–Pt/SiC. In reverse, the BE of Ir 4f for $Cu_2O$–Pt/SiC/$IrO_x$ is slightly positively shifted as compared with Pt/SiC/$IrO_x$. The above results show the existence of the strong interfacial interaction between co-catalyst and SiC, which would be favourable for the electron migration and transfer in $Cu_2O$–Pt/SiC/$IrO_x$.

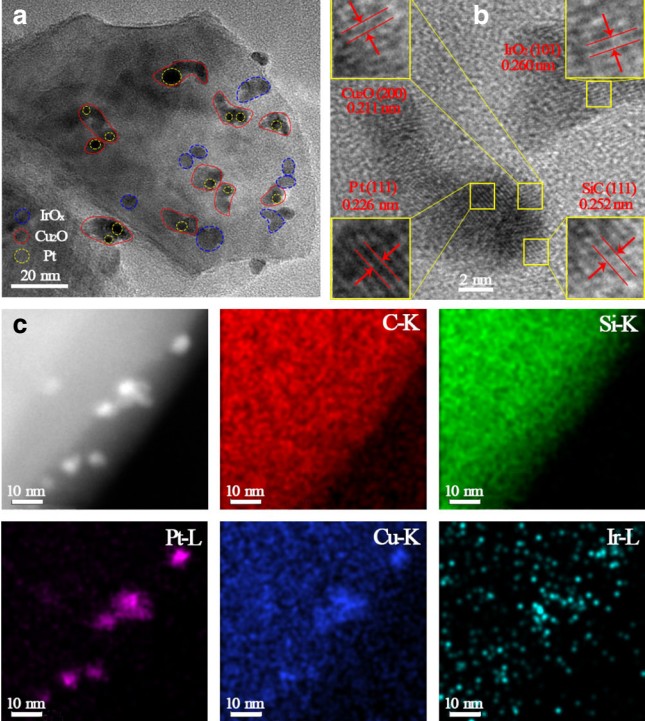

**Fig. 2 Spatial location of cocatalysts. a** TEM, **b** HRTEM images and **c** STEM image and corresponding EDS mapping profiles for C-K, Si-K, Pt-L, Cu-K and Ir-L of $Cu_2O$–Pt/SiC/$IrO_x$.

**Photocatalytic performance of $CO_2$ reduction with $H_2O$.** The photocatalytic $CO_2$ reduction was performed in the spatially separated reaction system. $Fe^{2+}$ and $Fe^{3+}$ were added respectively in the $CO_2$-reduction compartment and the $H_2O$-oxidation compartment in the beginning of the reaction (see Supplementary Fig. 6)[42,43]. During the photoreaction process, $Fe^{3+}$ and $Fe^{2+}$

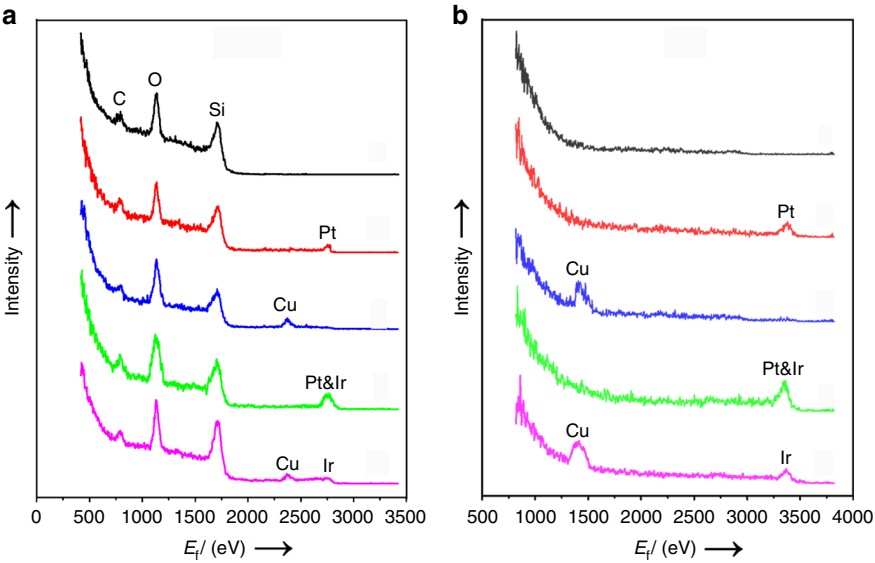

**Fig. 3 The chemical composition on the outermost surface.** HS-LEIS spectra using **a** 3 keV $^4He^+$ and **b** 5 keV $^{20}Ne^+$ for the samples: SiC (black), Pt/SiC (red), $Cu_2O$–Pt/SiC (blue), Pt/SiC/$IrO_x$ (green) and $Cu_2O$–Pt/SiC/$IrO_x$ (pink).

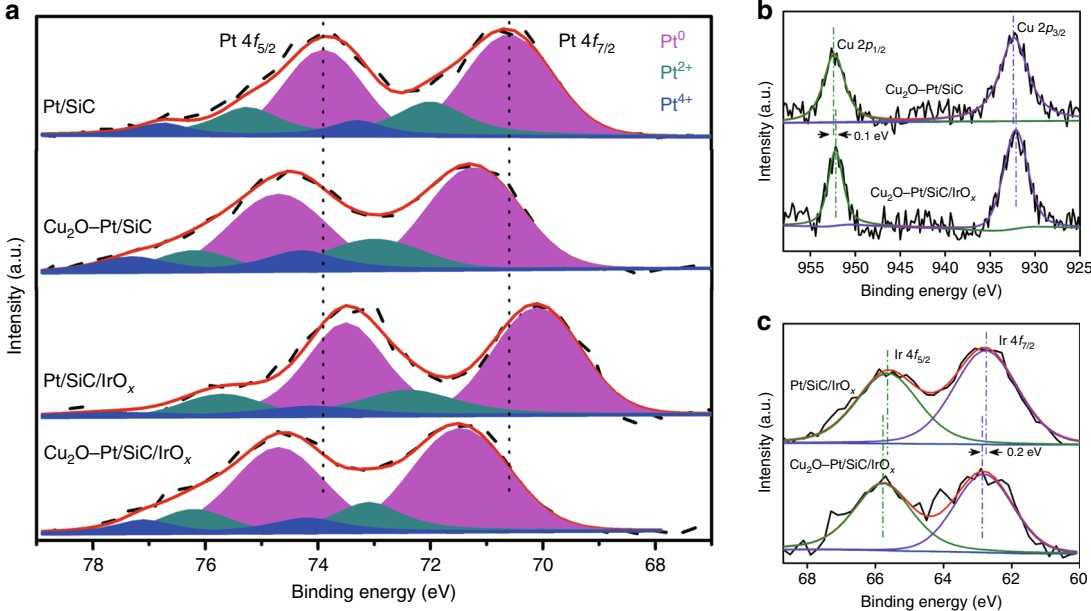

**Fig. 4 The chemical states of cocatalysts. a** Pt 4$f$ XPS spectra, **b** Cu 2$p$ XPS spectra, and **c** Ir 4$f$ XPS spectra of Pt/SiC, $Cu_2O$–Pt/SiC, Pt/SiC/$IrO_x$ and $Cu_2O$–Pt/SiC/$IrO_x$ samples.

ions are able to permeate through the Nafion membrane driven by the concentration gradient. The photocatalytic activities of the samples for the reaction of $CO_2$ with $H_2O$ were tested under visible light irradiation in order to find out the optimal contents of cocatalysts, as shown in Fig. 5 and Supplementary Table 1. HCOOH is detected as a main product for all photocatalysts. The bare SiC only shows 24.1 µmol $g^{-1}$ $h^{-1}$ of HCOOH yield. Over Pt/SiC samples, the HCOOH yield shows a volcanic curve with increasing Pt contents and the highest HCOOH yield (57.7 µmol $g^{-1}$ $h^{-1}$) occurs at 1.3 wt% Pt content (Fig. 5a). On the $Cu_2O$–Pt/SiC sample with 1.3 wt% Pt, the product HCOOH shows the highest yield (304.6 µmol $g^{-1}$ $h^{-1}$) at 1.8 wt% content of $Cu_2O$ (Fig. 5b). On the Pt/SiC/$IrO_x$ with the optimum Pt content (1.3 wt%), the optimal loading content of $IrO_x$ is 2.2 wt% at which the HCOOH yield is ca. 472.0 µmol $g^{-1}$ $h^{-1}$ (Fig. 5c). When both $IrO_x$ and $Cu_2O$ are simultaneously

deposited on the optimal Pt/SiC, the highest yield of HCOOH, 896.7 µmol $g^{-1}$ $h^{-1}$, occurs at ca. 2.2 wt% of $IrO_x$ and 1.8 wt% of $Cu_2O$. When $Cu_2O$ content is higher than 1.8 wt%, further increasing $Cu_2O$ photodeposition induces more amount of $Cu_2O$ on both Pt particles and the exposed SiC surface. The deposited $Cu_2O$ on SiC could block the optical absorption of SiC, and an over-thick $Cu_2O$ layer on Pt surface is also unfavourable to $CO_2$ reduction on $Cu_2O$–Pt[32]. The higher $IrO_x$ contents <2.2 wt%, the more active sites of $IrO_x$ are provided to enhance the $Fe^{2+}$ oxidation. However, an excess amount of $IrO_x$ may lead to the growth of $IrO_x$ into large particles on SiC surface, which is harmful to the photocatalytic reaction. The HCOOH yield over the optimal $Cu_2O$–Pt/SiC/$IrO_x$ is almost 37 times of the activity of the bare SiC. To the best of our knowledge, the photocatalytic efficiency of our system is substantially higher than that of other various photocatalysts

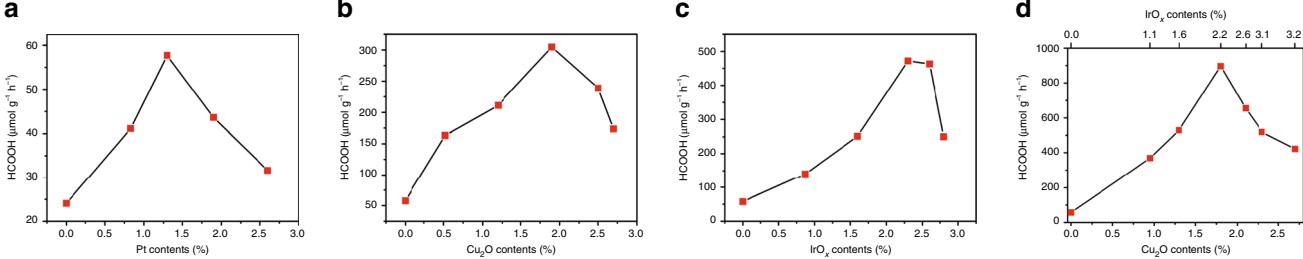

**Fig. 5 The optimal contents of cocatalysts for CO₂ reduction.** Change in HCOOH evolution rate **a** on Pt/SiC with increasing Pt content, **b** on Cu₂O–Pt (1.3 wt%)/SiC with increasing Cu₂O, **c** on Pt (1.3 wt%)/SiC/IrO$_x$ with increasing IrO$_x$, and **d** on Cu₂O-Pt/SiC/IrO$_x$ with increasing Cu₂O and IrO$_x$, in spatially separated reactor under visible light irradiation. Reaction conditions: 50 mg SiC-based photocatalyst and 300 mL of 2 mM FeCl₂ aqueous solution were placed in the CO₂-reduction chamber, 100 mg Pt/WO₃ and 300 mL of 2 mM FeCl₃ aqueous solution in the H₂O oxidation chamber, the pH of solution was adjusted to 2.3 by adding hydrochloric acid to prevent precipitation of the iron ions aqueous solution.

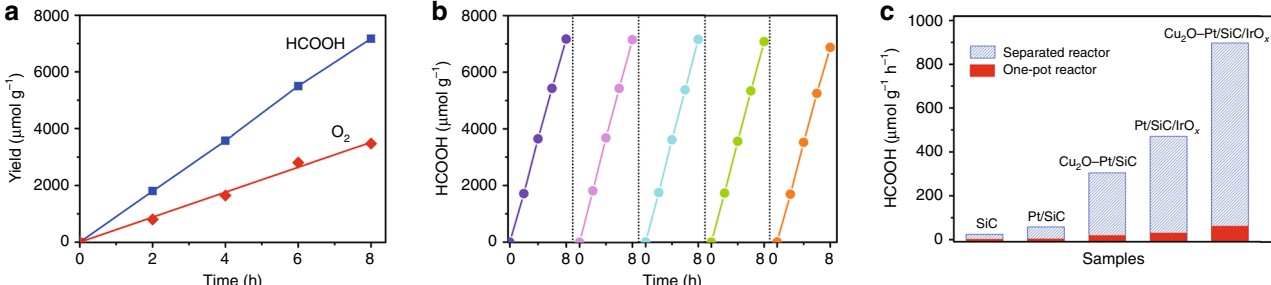

**Fig. 6 Photocatalytic performance of CO₂ reduction with H₂O. a** Evolutions of HCOOH and O₂ as a function of illumination time and **b** cycle experiment of HCOOH evolution in the spatially separated reaction system with Cu₂O-Pt/SiC/IrO$_x$ as the reduction side photocatalyst. **c** Comparison for the HCOOH evolution between in the spatially separated reaction system and in the conventional one-pot reactor with different samples as the reduction side photocatalyst.

reported so far (seen activity comparison in Supplementary Table 3).

O₂ is confirmed as main product evolving in the Pt/WO₃ oxidation chamber of the separated reaction system. Figure 6a shows the production of O₂ in the Pt/WO₃ chamber and HCOOH in the Cu₂O–Pt/SiC/IrO$_x$ chamber vs. irradiation time. Both yields of HCOOH and O₂ increase linearly, and reach 7200 and 3300 μmol g$^{-1}$ during 8 h photocatalytic reactions, respectively. Moreover, the ratio of product O₂ to HCOOH is close to the stoichiometric number of reaction $(2CO_2 + 2H_2O \rightarrow 2HCOOH + O_2)$ in the whole process. Influence of the Pt amount of Pt/WO₃ on the photocatalytic O₂ and HCOOH evolutions in the separated reaction system was also investigated (Supplementary Fig. 7). As decreasing Pt content to 0.5 wt% of Pt/WO₃, the O₂ and HCOOH yields decrease to 296.6 and 618.7 μmol g$^{-1}$ h$^{-1}$, respectively. However, the ratio of O₂ to HCOOH is still close to the stoichiometric number. When the Pt content increases to 1.5 wt%, both O₂ and HCOOH evolution is almost the same as that of 1.0 wt% Pt. The 1.0 wt% Pt content is optimal for the Pt/WO₃ in the separated reaction system. Obviously, Pt content of Pt/WO₃ does not affect the stoichiometric ratio of product HCOOH to O₂. This is because Fe³⁺/Fe²⁺ redox couple acts as the electron transfer medium between CO₂ reduction and H₂O oxidation by the concentration diffusion through the Nafion membrane, which makes the reaction stoichiometrically proceeded in the reaction process. However, the HCOOH yield shows a downtrend as prolonging the reaction time (Supplementary Fig. 8). The decrease of activity is because pH value of the reaction solution is increased from 2.3 at the initial stage to 4.5 after 12 h reaction along with the consumption of H⁺ in the process of CO₂ reduction. The increase in solution alkalinity can result in hydrolysis of Fe³⁺ to FeOOH or Fe(OH)₃, as confirmed by the Fe 2p XPS spectrum of the sample after the reaction

(Supplementary Fig. 9). However, the activity is highly stable in five cycles of total 40 h (Fig. 6b) when the Cu₂O–Pt/SiC/IrO$_x$ is centrifuged out and then added to the fresh solution before each new cycle. Therefore, the increase in solution alkalinity during the photocatalytic reaction is one of reasons effecting stability of the reaction, which should be avoided by increasing solution acidity.

The Cu₂O–Pt/SiC/IrO$_x$ samples before and after the photocatalytic reaction were characterized by XPS (Supplementary Fig. 10), indicating some differences in chemical states of Pt, Ir and Cu. However, the changes of metal states are likely to have no significantly influence or be not main factor on the photocatalytic activity stability. (i) For Pt, the XPS Pt 4f peaks narrow down and the shoulder peaks become weak for the used sample, but the BE values of main peaks keep almost unchanged before and after the photocatalytic reaction. The main peaks at the BEs of 74.7 eV (Pt 4f 5/2) and 71.5 eV (Pt 4f 7/2) are attributed to Pt(0). The shoulder peaks are belonged to Pt²⁺ and Pt⁴⁺ species. It can be seen that a part of high valence Pt species were translated also into Pt (0) after longtime reaction, which could be beneficial to the photocatalysis. (ii) For IrO$_x$, the Ir 4f peaks not only become narrow but also shift towards lower energy after the photocatalytic reaction. The wide Ir 4f peaks of both the fresh and the used samples cover Ir⁰, Ir³⁺ and Ir⁴⁺ species, indicating the mix valence state feature of IrO$_x$. Accordingly, the Ir 4f$_{7/2}$ peak can be deconvoluted into a contribution of Ir⁰ (61.3 eV), Ir⁴⁺ (62.4 eV) and Ir³⁺ (63.5 eV) species[44–46]. The phenomena are universal for Ir-loaded catalysts, which is also the reason why iridium oxide is usually be expressed as IrO$_x$ rather than IrO₂[47]. It is estimated from the peak intensities that the contents of Ir⁴⁺ are increased, while Ir³⁺ contents are decreased in the used sample as compared with the fresh sample (see Supplementary Table 4). Such a change in the Ir 4f XPS spectra can be explained by the changes in the crystallinity and coordination numbers[48,49]. For the fresh sample,

the broader and higher BE peaks suggest the existence of partial amorphous or high oxygen coordinated Ir species. For the used sample, the shift of BE of Ir 4$f$ peak towards lower energy indicates an increase in the rutile phase $IrO_2$ during the photocatalytic process since the $IrO_x$ could be mainly excited from the d($t_{2g}$) to the d($e_g$) band (1.5–2.75 eV) under visible light irradiation based on the literature[50]. It is possible that the change in Ir valence state or crystallinity do not affect the photoinduced d–d transition and thus photocatalytic performance. (iii) As for $Cu_2O$, the Cu 2$p$ XPS spectrum shows a minor change. The BEs of the main peaks of Cu 2$p$ remain almost the same before and after the reaction, only a minuscule shift towards lower energy. We could not exclude the possibility that a small amount of Cu(I) was translated into Cu(0) after the photocatalytic reaction. Because some $Cu_2O$ is deposited synchronously on the SiC surface, the small change of Cu valence state can occur partly for these $Cu_2O$ species, which could have no remarkable influence on the photocatalytic performance.

Other two control experiments were carried out also. When $^{13}CO_2$ instead of $CO_2$ was used as a reactant, $^{13}C$ NMR analysis for the reaction solution verifies that only a strong peak at 171.5 ppm attributed to the $^{13}C$ in $H^{13}COOH$ was observed (Supplementary Fig. 11)[51]. When $CO_2$ was not added in the reaction system, no HCOOH was detected. These results show that the product HCOOH is formed from the photocatalytic $CO_2$ reduction. As $H_2^{18}O$ instead of $H_2O$ is used as the reactant in a small dose simulated reaction, the mass spectroscopy analysis gives a main peak at $m/z = 36$ corresponding to $^{18}O_2$ (Supplementary Fig. 12), confirming the product $O_2$ originating from $H_2O$ oxidation. Careful analysis reveals that except HCOOH no carbon-containing products, such as CO, HCHO and $CH_4$, come into being in the gas phase. Nevertheless, there is only a very slight amount of $H_2$ to evolve as a by-product (Supplementary Fig. 13), implying existence of competition between the reduction of $H^+$ to $H_2$ and the reduction of $CO_2$ to HCOOH by photogenerated electrons. In the absence of $CO_2$ atmosphere, however, only a slightly enhanced amount of $H_2$ evolves in the gas phase (Supplementary Fig. 14), although the photocatalytic reaction is performed in the acidic aqueous solution. This indicates that the SiC-based photocatalyst is not good for $H_2$ evolution from water, which may be because that it lacks the active sites or has high overpotential for the $H_2$ evolution. However, for the reduction of $CO_2$, the SiC-based photocatalyst can be particularly effective.

The photocatalytic activity for the reaction of $CO_2$ with $H_2O$ was also evaluated in the conventional one-pot reactor, where we added simultaneously SiC-based catalyst, $Pt/WO_3$ and $Fe^{2+}/Fe^{3+}$. HCOOH and $O_2$ are also detected as main products. Figure 6c compares the HCOOH yields under two different reaction modes (see also Supplementary Table 5). It can be seen that all the samples show much higher HCOOH yield with the spatially separated reaction system than with the one-pot reaction mode. For SiC, Pt/SiC, $Cu_2O$–Pt/SiC, $Pt/SiC/IrO_x$ and $Cu_2O$–Pt/SiC/$IrO_x$ photocatalyst, the HCOOH yield is 1.7, 3.5, 18.6, 30.4 and 61.5 μmol g$^{-1}$ h$^{-1}$ with the one-pot reaction, while it is 24.1, 57.7, 304.6, 472.0 and 896.6 μmol g$^{-1}$ h$^{-1}$ with the spatially separated system, respectively. Obviously, the photocatalytic activity of the spatially separated system is ca ~15 times higher than that in the one-pot reaction reactor for each photocatalyst. In the one-pot reaction system, the backward reaction of HCOOH re-oxidization by $O_2$ should be one of the reasons for the low evolution of HCOOH and $O_2$. Both $Fe^{3+}$ competing with $CO_2$ for the generated electrons and $Fe^{2+}$ competing with $H_2O$ for the photogenerated holes could also take place at the same time. However, the effects of the later could be weaker than that of the

former, because the evolutions of HCOOH and $O_2$ are lower if no adding $Fe^{3+}$ and $Fe^{2+}$ (see the first column in Supplementary Table 5). This could be one of reasons for high efficiency of the spatially separated system. However, the reasonableness of this inference requires the following two premises: the product HCOOH (i) does not diffuse to the oxygen evolution chamber from the reduction chamber through the Nafion membrane and (ii) is not oxidized in the reduction chamber by the $Fe^{3+}$, with increase in the production of HCOOH. Accordingly, two additional experiments were done. The permeation of HCOOH across the Nafion membrane from the HCOOH solution (1200 μmol L$^{-1}$, corresponding to the maximum yield of HCOOH in the separated system for 8 h reaction) to pure $H_2O$ is determined firstly. Only very limited amount of HCOOH (<5%) is diffused to the pure water across the Nafion membrane within 8 h (Supplementary Fig. 15). Another experiment is the solution (pH = 2.3) with HCOOH (1200 μmol L$^{-1}$) and $Fe^{2+}/Fe^{3+}$ (2 mmol L$^{-1}$) under visible light illumination. The concentration of HCOOH remains almost unchanged (Supplementary Fig. 16). These indicate that our new system is indeed effective to prevent the backward reaction. The wavelength-dependent evolution of HCOOH was also performed to gain the apparent quantum yield (AQY) (Supplementary Fig. 17). Obviously, the AQY of HCOOH evolution on the optimal $Cu_2O$–Pt/SiC/$IrO_x$ sample is well matched with the optical absorption spectra. Under 400 nm light irradiation, the AQY of HCOOH evolution can be reached near 1.44%.

## The in situ $CO_2$ adsorption FT-IR spectra

The photocatalytic activity of $Cu_2O$–Pt/SiC/$IrO_x$ is always higher than that of the other samples both in the spatially separated system and in the one-pot reactor. The HCOOH yield on the optimal $Cu_2O$–Pt/SiC/$IrO_x$ sample in the spatially separated system is 37 and 527 times higher than that on the bare SiC in the spatially separated system and in the one-pot reactor, respectively. Obviously, the high efficiency of $Cu_2O$–Pt/SiC/$IrO_x$ for the reaction of $CO_2$ with $H_2O$ to HCOOH can be related to the photocatalyst surface feature. The in situ $CO_2$ adsorption FT-IR spectra were measured to gain insight into the effect of surface feature. As shown in Fig. 7a, all photocatalysts show the multiple IR adsorption peaks of $CO_2$ in the range of 2200–2500 cm$^{-1}$ in the dark. It is noteworthy that both the samples containing Cu, i.e. $Cu_2O$–Pt/SiC and $Cu_2O$–Pt/SiC/$IrO_x$, show much stronger $CO_2$ adsorption band than nude SiC, Pt/SiC and Pt/SiC/$IrO_x$. This implies that $CO_2$ molecules are mainly adsorbed on the $Cu_2O$ co-catalyst in $Cu_2O$–Pt/SiC and $Cu_2O$–Pt/SiC/$IrO_x$[32,52]. Figure 7b shows the in situ $CO_2$ adsorption FT-IR spectra on $Cu_2O$–Pt/SiC/$IrO_x$ in the range of 1700–1200 cm$^{-1}$ after and before light irradiation, which is a reflection of chemical adsorption. No absorption attributable to C–O species appears on $Cu_2O$–Pt/SiC/$IrO_x$ in the absence of $CO_2$ gas (omitted). As introduction of $CO_2$, $Cu_2O$–Pt/SiC/$IrO_x$ shows very weak several absorption peaks before visible light irradiation. However, three strong absorption bands appear upon visible light irradiation. Both the wide absorption band at 1394 cm$^{-1}$ and the weak band at 1503 cm$^{-1}$ are ascribed to bidentate carbonate species bonded to the $Cu_2O$ surface, while the absorption band at 1262 cm$^{-1}$ is assigned to a monodentate carbonate to the $Cu_2O$ surface[53,54]. This demonstrates that $CO_2$ molecules are activated at room temperature by the $Cu_2O$ co-catalyst on $Cu_2O$–Pt/SiC/$IrO_x$ surface under light irradiation. Moreover, the acidic aqueous solution (pH = 2.3) is helpful for formation of the carboxyl radical intermediate (·COOH) or the formate anion ($HCOO^-$) that is further easily converted into HCOOH[14,55]. This could be the second possible reason for the high efficiency of the spatially separated system. High selectivity of product HCOOH may be related to the different reaction mechanism dependent on the

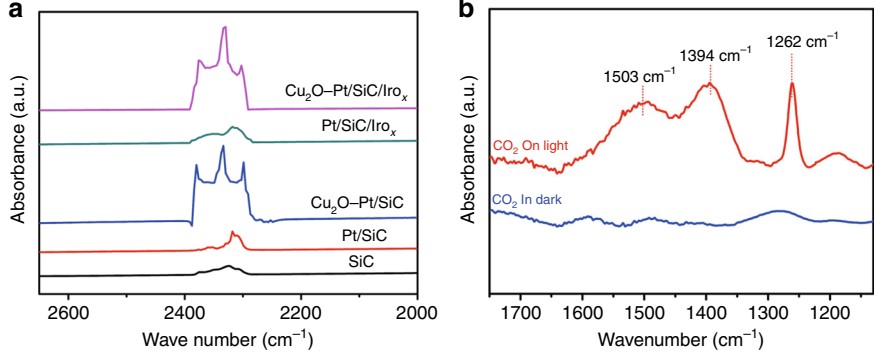

**Fig. 7 Adsorption of CO₂ on photocatalysts. a** In situ FT-IR spectra of $CO_2$ adsorbed on different photocatalysts. **b** In situ FT-IR spectra of $CO_2$ adsorbed on Cu₂O–Pt/SiC/IrO$_x$ before and after visible light irradiation. All the spectra are the difference spectra between after and before $CO_2$ adsorption.

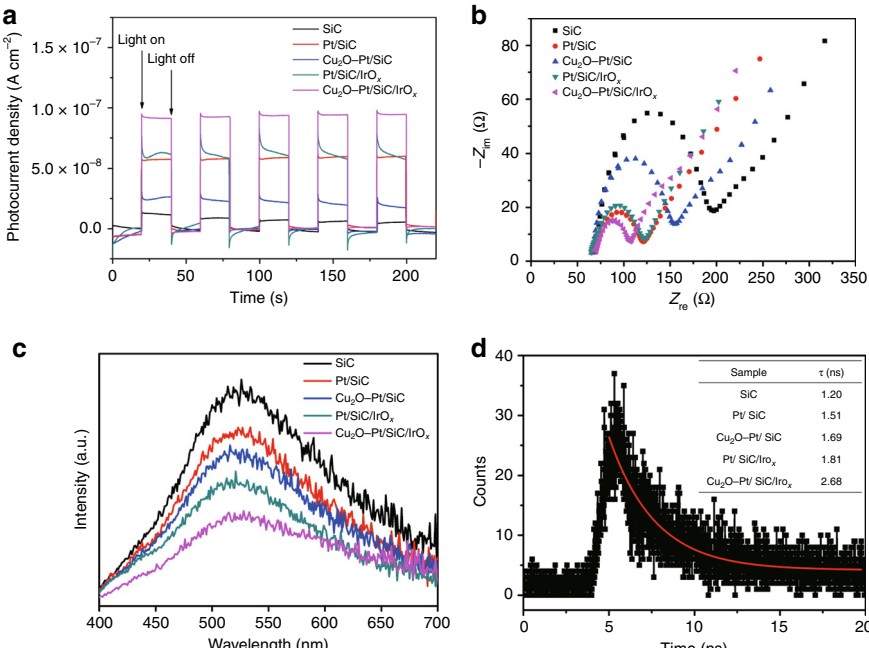

**Fig. 8 Separation efficiency and lifetime of carriers. a** Periodic on/off photocurrent response, **b** AC Impedance and **c** PL (photoluminescence) spectra for different photocatalysts. **d** Time-resolved photoluminescence spectroscopy of Cu₂O-Pt/SiC/IrO$_x$.

reaction conditions, such as the photocatalyst states (defects, crystal faces, doping, etc.), reaction conditions (temperature, pH, $CO_2$ concentration, reactant phase, etc.), co-catalyst and so on. In the gas ($CO_2$, $H_2O$ vapour)–solid (catalyst.) mode, it has been found that the SiC-based composite ($MoS_2$/SiC) photocatalyzed the $CO_2$ reduction into $CH_4$ undergoing $HCOOH$, $HCHO$ and $CH_3OH$ intermediates on SiC surface by the hydrogenation pathways[26]. In the present work, the reaction is conducted in gas ($CO_2$)–liquid ($H_2O$)–solid(catalyst) mode and the acidic aqueous solution. In addition, the reduction of $CO_2$ occurs on the $Cu_2O$ sites rather than SiC, which is helpful for formation of the carboxyl (hydroxyformyl) radical intermediate ·COOH and $HCOOH$[14,56].

**Photoelectrochemical properties of Cu₂O–Pt/SiC/IrO$_x$.** The photoelectrochemical responses and the photoluminescence spectra of these photocatalysts were measured. Figure 8a, b shows the photocurrents and AC impedance of the different samples. The photocurrents increase in order of SiC, Cu₂O–Pt/SiC, Pt/SiC, Pt/SiC/IrO$_x$ and Cu₂O–Pt/SiC/IrO$_x$, and the AC impedance values decrease in the same order. Obviously, Cu₂O–Pt/SiC/IrO$_x$

shows the biggest photocurrent and the smallest AC impedance, indicating the highest electron transfer rate and separation efficiency. However, the sample Cu₂O–Pt/SiC and Pt/SiC show a slight abnormality, i.e. both the photocurrent increase order and the AC impedance decrease order are inconsistent with the increase order of photocatalytic activity. This can be explained by the dependence of photocatalytic activity on not only the transfer rate and separation efficiency of photogenerated charges but also the surface chemistry characteristics of photocatalyst. Figure 8c is the steady-state photoluminescence (PL) spectra of these samples. The PL intensities decrease in accordance with the order of SiC, Pt/SiC, Cu₂O–Pt/SiC, Pt/SiC/IrO$_x$ and Cu₂O–Pt/SiC/IrO$_x$, which agree with the change in the photocatalytic activity. Since low PL intensity corresponds to low recombination rate of photogenerated charges, the lowest PL intensity indicates the smallest recombination rate of photogenerated charge for Cu₂O–Pt/SiC/IrO$_x$. Figure 8d shows the time-resolved photoluminescence spectroscopy of these samples (see detailed Supplementary Fig. 18), and the corresponding average lifetimes ($\tau$) of charge carriers (the inset table). The $\tau$ value of nude SiC is about 1.2 ns, consistent with the reported literature[57]. Loading Pt, Cu₂O and

$IrO_x$ nanoparticles lead to an increase in the lifetime $\tau$. In these samples, $Cu_2O$–Pt/SiC/$IrO_x$ has the highest $\tau$ value. The increase in the lifetime $\tau$ of the charge carriers could increase the probability of their involvement in photocatalytic reactions before recombination[58–60]. So, increases in the transfer rate, separation efficiency and lifetime of photogenerated charges could be the third reason for high efficiency of the spatially separated system.

**Photocatalytic mechanism of the spatially separated system.** The next issue is the electron structure of $Cu_2O$–Pt/SiC/$IrO_x$ and the photocatalytic mechanism of the spatially separated system. The electron structure parameters of SiC and $Cu_2O$ were reckoned by combination of the UV–Vis diffuse reflection spectra with the Mott-Schottky analysis (Supplementary Fig. 19). The optical absorption edge of the nude SiC is at 501 nm. The band gap energy and the conduction band (CB) potential of SiC are ca. 2.48 eV and −1.08 V (vs. SHE), respectively. The parent $Cu_2O$ is estimated to have a band gap of 1.98 eV and CB of −1.28 V. It has been reported that the $IrO_x$ could be excited from the d($t_{2g}$) to the d($e_g$) band (1.5–2.75 eV) by visible irradiation and from the O-p band to the d($e_g$) (>3.0 eV) band by ultraviolet irradiation, and its CB is +0.35 V[50]. For the $Cu_2O$–Pt/SiC/$IrO_x$ photocatalyst, $Cu_2O$–Pt and $IrO_x$ cocatalysts are separated each other on SiC surface, and Pt is sandwiched between $Cu_2O$ and SiC. Thus, $Cu_2O$–Pt/SiC/$IrO_x$ is suggested to have the energy band alignment in Fig. 9a.

If the $Cu_2O$ co-catalyst directly contacts with SiC without Pt, the photoelectron transfers on the resulting $Cu_2O$/SiC composite is speculated to follow the Z-scheme transfer from SiC to $Cu_2O$, which has been reported in many literatures[61–63]. Such Z-scheme photoelectron transfer was verified by a photocatalytic probe reaction. When the water solution containing the $Cu_2O$/SiC sample and $H_2PtCl_6$ was illuminated by UV light, the reduction reaction ($PtCl_6^{2-} + 4e^- \rightarrow Pt + 6Cl^-$) would occur on the $Cu_2O$/SiC surface. The TEM image clearly indicates preponderant photodeposition of Pt particles over $Cu_2O$ particle rather than over SiC surface (Supplementary Fig. 20), unambiguously verifying the Z-scheme electron transfer, i.e. the transfer of photogenerated electron from the CB of SiC to the VB of $Cu_2O$. For our $Cu_2O$–Pt/SiC/$IrO_x$ sample, we can reasonably conclude that the Z-scheme electron transfers from the CB of SiC to the valence band (VB) of $Cu_2O$ would be accelerated by the Pt nanoparticles located between $Cu_2O$ and SiC, due to the excellent conductivity and high work function of Pt. The controlled experiment shows that the $Cu_2O$/SiC displays the HCOOH evolution of about 40.5 $\mu mol\,g^{-1}\,h^{-1}$, which is little higher than that of the pristine SiC but much lower than that of $Cu_2O$–Pt/SiC

samples (Supplementary Fig. 21). This means that contribution of the $Cu_2O$ deposited on SiC surface to the activity is very small, and the high activity is mainly due to the embedding of Pt in the interface between $Cu_2O$ and SiC. The photocatalytic $CO_2$ reduction performances of $Cu_2O$/SiC/$IrO_x$ and Pt–$Cu_2O$/SiC/$IrO_x$ were also compared with that of $Cu_2O$–Pt/SiC/$IrO_x$ (Supplementary Fig. 21). Much lower activities of $Cu_2O$/SiC/$IrO_x$ and Pt–$Cu_2O$/SiC/$IrO_x$ than that of $Cu_2O$–Pt/SiC/$IrO_x$ indicates likewise that the Pt sandwiched between the interface of $Cu_2O$ and SiC is more beneficial to the transfer of photogenerated electrons from SiC to $Cu_2O$ thus enhances $CO_2$ reduction. Simultaneously, another direct Z-scheme photoelectron transfer occurs at the interface between $IrO_x$ and SiC, because SiC has much more positive VB than CB of $IrO_x$. As a result, SiC, $Cu_2O$ and $IrO_x$ in $Cu_2O$–Pt/SiC/$IrO_x$ all are excited by visible light, the photogenerated electrons from the CB of $IrO_x$ (+0.35 V) would transfer towards the VB of SiC (+1.40 V). Synchronously, the photogenerated electrons from the CB of SiC (−1.08 V) would transfer towards Pt nanoparticles, and then towards the VB of $Cu_2O$ (+0.70 V) where they combine with the photogenerated holes at $Cu_2O$. On the whole, the coupled direct Z-scheme processes result in the photogenerated electrons accumulating in the CB of $Cu_2O$ (−1.28 V) where the adsorbed $CO_2$ is reduced into HCOOH [$E(CO_2/HCOOH) = -0.61$ V], while the photogenerated holes on the VB of $IrO_x$ ($t_{2g} = +1.85$ V) to oxidize $Fe^{2+}$ into $Fe^{3+}$ [$E(Fe^{2+}/Fe^{3+}) = +0.77$ V][64,65]. Meanwhile, in the $H_2O$ oxidation chamber, Pt/$WO_3$ is excited also by visible irradiation. The photogenerated electrons would transfer from the CB of $WO_3$ (+0.74 V) towards Pt and then reduce $Fe^{3+}$ into $Fe^{2+}$ [$E(Fe^{2+}/Fe^{3+}) = +0.77$ V], while the photogenerated holes of $WO_3$ (+2.06 V) would oxidize $H_2O$ to $O_2$ ($E = +1.23$ V). As shown in Fig. 9b, the overall photocatalytic system follows indirect Z-scheme mechanism similar to the natural photosynthesis, i.e. the photocatalytic $CO_2$ reduction half-reaction and the photocatalytic $H_2O$ oxidation half-reaction take place at two separated reactors by the relaying role of $Fe^{3+}$/$Fe^{2+}$ redox couple. To prove that the total reaction is a combination of the spatially separated reduction and oxidation, two controlled experiments were done. When the water solution containing photocatalyst Pt/$WO_3$ and $Fe^{3+}$ is illuminated with visible light, $O_2$ is generated also as a main product (Supplementary Fig. 22a), validating the oxidation half-reaction in the left side of Fig. 9b. When the solution containing $H_2O$, $Cu_2O$–Pt/SiC/$IrO_x$, $Fe^{2+}$ and $CO_2$ is illuminated with visible light, HCOOH is detected also as main product (Supplementary Fig. 22b), validating the reduction half-reaction in the right side of Fig. 9b. Therefore, $Fe^{3+}$/$Fe^{2+}$ redox couple makes the reduction of $CO_2$ to HCOOH on

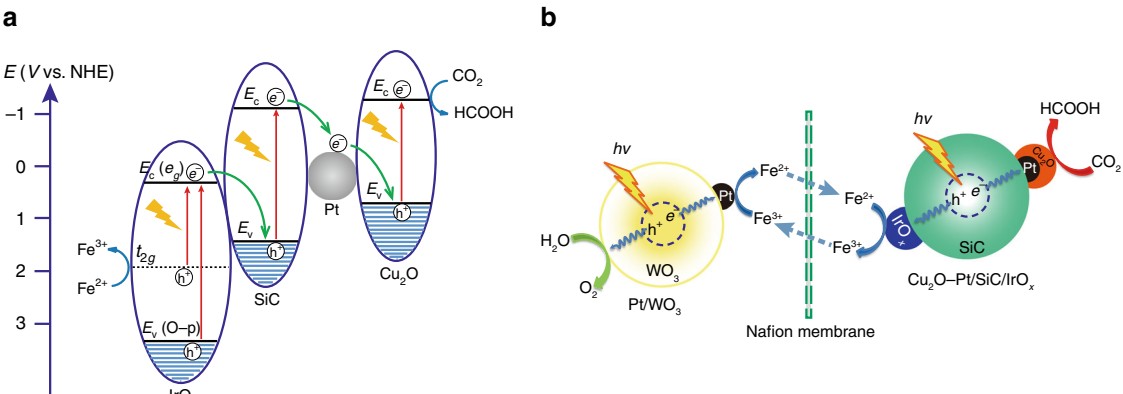

**Fig. 9 Electronic structure and photocatalytic mechanism. a** The electron transfer processes in $Cu_2O$–Pt/SiC/$IrO_x$ under light illumination. **b** The proposed mechanism of the separated system for the efficient $CO_2$ reduction and $O_2$ evolution.

$Cu_2O$–Pt/SiC/$IrO_x$ and the oxidation of $H_2O$ to $O_2$ on Pt/$WO_3$ integrated into the one system like the natural photosynthetic systems. It must be mentioned that the usage mode of $Fe^{3+}$ and $Fe^{2+}$ has little impact on the reaction efficiency. When the $Fe^{2+}$/$Fe^{3+}$ mixed solution was used in both cells of the separated reaction system, HCOOH was produced steadily with the reaction time (Supplementary Fig. 23), and its generation rate is 629.8 $\mu mol\,g^{-1}\,h^{-1}$. When $Fe^{2+}$ and $Fe^{3+}$ were separately loaded in the $CO_2$-reduction compartment and the $H_2O$-oxidation compartment, the initial generation rate of HCOOH is 896.6 $\mu mol\,g^{-1}\,h^{-1}$. Obviously, the separate addition of $Fe^{2+}$ solution and $Fe^{3+}$ solution in both compartments is more efficient than the loading of $Fe^{2+}$/$Fe^{3+}$ mixed solution in both compartments for the photocatalytic reaction in the separated reaction system. But the separated use mode of $Fe^{2+}$ and $Fe^{3+}$ can only improve initiate photoreaction efficiency. When the redox reaction reaches dynamic steady state, the concentrations of $Fe^{2+}$ and $Fe^{3+}$ also reach a constant gradient in both sides to maintain counter diffusion. $Fe^{2+}$ and $Fe^{3+}$ are not evenly dispersed in either cell according to previous study due to osmosis of Fe ions through Nafion membrane[66]. Even though the concentration of $Fe^{2+}$ and $Fe^{3+}$ is equal in the reaction compartments, the $H_2O$ oxidation and $CO_2$ reduction can proceed over the photocatalysts due to the different adsorption property of $Fe^{2+}$/$Fe^{3+}$ on photocatalyst surface. For example, $Fe^{3+}$ is more favourable to adsorb on $WO_3$ surface than $Fe^{2+}$ (ref. [67]).

## Methods

**Materials**. $SiO_2$ and $H_2PtCl_6\cdot6H_2O$ were acquired from Aladdin, and the $Na_3IrCl_6$ was supplied by Alfa Aesar. Other reagents used in this work, including glucose, methanol, $WO_3$, NaOH, $CuSO_4\cdot5H_2O$ and $NaIO_3$ were of analytical reagent grade and obtained from Sinopharm Chemical Reagent Co., Ltd. All of the above chemicals were used without further purification.

**Preparation of Pt/SiC photocatalyst**. The SiC nanoparticles were prepared by the carbothermic reduction method. Powders of $SiO_2$ and glucose were mixed in the molar ratio of Si:C = 1:6 and pulverized in a mortar to well disperse the mixed powders. The well-mixed powder was calcined under Ar atmosphere at 1450 °C for 5 h at a rate of 2 °C min$^{-1}$ in a tubular furnace. The calcined powder was then cooled to room temperature and further purified to remove unreacted raw materials based on our previous report[26]. Briefly, the sample was calcined under $O_2$ atmosphere at 873 K for 5 h and steeped with 10 wt% sodium hydroxide solution in order to remove free carbon and unreacted $SiO_2$.

Platinum (Pt) was loaded on SiC by the photodeposition method as described in the following steps. The $H_2PtCl_6$ solution with a concentration of 1 mmol L$^{-1}$ was mixed with SiC powder in a quartz cell. In the process of stirring, 2 mL of methanol was added into the mixture. After evacuation, the suspension was irradiated with a 125 W Hg lamp to load Pt nanoparticles onto the SiC. After a certain irradiation time, the obtained sample was washed thoroughly with deionized water and dried in vacuum for 1 h. The irradiation time was changed from 0.5 to 2 h to tune the content of Pt in sample, which was denoted as Pt−$x$h/SiC ($x$ = 0.5, 1, 1.5, 2, representing the irradiation time).

**Preparation of $Cu_2O$–Pt/SiC and Pt/SiC/$IrO_x$ photocatalysts**. $Cu_2O$–Pt/SiC and Pt/SiC/$IrO_x$ photocatalysts were also prepared by the similar procedure by using Pt/SiC (Pt-1h/SiC) instead of pure SiC. In the preparation of $Cu_2O$–Pt/SiC, $CuSO_4\cdot5H_2O$ aqueous solution in the concentration of 0.6 mmol L$^{-1}$ and 2 mL of methanol were introduced to the quartz cell together with Pt/SiC. The content of Cu species in the final sample was controlled by changing the irradiation time from 1 to 10 h, the resulting sample was denoted as $Cu_2O$–Pt−$y$h/SiC ($y$ = 1, 3, 5, 8, 10). For the synthesis of Pt/SiC/$IrO_x$, $Na_3IrCl_6$ and $NaIO_3$ aqueous solution with the respective concentration of 0.6 and 0.01 mol L$^{-1}$ were added into the reactor to mix with Pt/SiC. The content of Ir species in the final sample was adjusted also by the irradiation time from 1 to 10 h, the resulting sample was denoted as Pt/SiC/$IrO_x$ − $y$h ($y$ = 1, 3, 5, 8, 10). After $IrO_x$ photodeposition, the samples were washed by deionized water and ethanol for several times to remove residual iodine species. The final sample was obtained after drying in vacuum for 1 h. In order to detect the residual iodine ions, the samples were immersed in the $AgNO_3$ solution. However, we did not find any precipitates $AgIO_3$ or AgI in solution, indicating iodine ion can be completely washed away in the synthesis process.

**Preparation of $Cu_2O$–Pt/SiC/$IrO_x$ photocatalysts**. Pt/SiC was added to the aqueous solution with $CuSO_4\cdot5H_2O$ (0.6 mmol L$^{-1}$) and $Na_3IrCl_6$ (0.6 mmol L$^{-1}$)

followed by evacuation and irradiating with a 125 W Hg lamp. The content of Cu and Ir species in the final sample was controlled by the irradiation time from 3 to 15 h. The obtained sample was washed thoroughly with deionized water and dried in vacuum for 1 h, which was denoted as $Cu_2O$–Pt-$z$h/SiC/$IrO_x$ ($z$ = 3, 5, 8, 10,12, 15).

**Preparation of Pt/$WO_3$ photocatalyst**. Pt/$WO_3$ photocatalyst was also prepared by the photoreduction method similar to that in preparation of Pt/SiC photocatalyst with the irradiation time of 0.5 h.

**Characterization of photocatalysts**. XRD patterns were recorded with Ni filtered Cu Kα radiation at 40 kV and 40 mA on a Bruker D8 Advance X-ray diffractometer. Morphology of sample was characterized by a field emission scanning electron microscopy (JSM-6700F) and TEM. TEM images were obtained at an accelerating voltage of 200 kV using a JEOL model JEM 2010 EX instrument. UV–Vis diffuse reflectance (UV–Vis DRS) spectra were obtained on a UV–Vis spectrophotometer (Cary 500) with a self-supporting sample cell, and the pure $BaSO_4$ was used as a reflectance standard. Brunauer–Emmett–Teller (BET) surface area was measured with an ASAP2020M apparatus (Micromeritics Instrument Corp., USA). Nitrogen adsorption and desorption isotherms were measured at 77 K. Contents of Pt, Cu and Ir in the samples were measured using an inductively coupled plasma optical emission spectrometer (Ultima 2, HORIBA Jobin Yvon Co., France). HS-LEIS measurements were carried out on an IonTOF Qtac100 low-energy ion scattering analyser. $^4He^+$ ions with a kinetic energy of 3 keV were applied at a low ion flux equal to 1325 pA cm$^{-2}$, which was necessary to avoid the sputtering of surfaces. $^{20}Ne^+$ ions with a kinetic energy of 5 keV were applied at a low ion flux equal to 445 pA cm$^{-2}$. The scattering angle was 145°. XPS measurements were performed on a Quantum 2000 Scanning ESCA Microprobe (Physical Electronics) using Al Kα radiation (1846.6 eV) as the X-ray source. $^{13}C$ NMR spectra were recorded on AVANCE III 400 MHz spectrometer using TMS as the internal standard. $^{13}CO_2$ and $H_2^{18}O$ were employed as the reactants in the isotope labelling comparison reaction instead of $CO_2$ and $H_2O$, respectively. After irradiation, the reaction solution was characterized with $^{13}C$ NMR directly. FT-IR experiments were carried out on a Nicolet 670 FT-IR spectrometer at a resolution of 4 cm$^{-1}$ and 64 scans. FT-IR experiments were performed in a home-made IR cell in conjunction with a vacuum system. The catalyst powders were first pressed into a self-supporting disk (18 mm diameter, 20 mg), and then the disk was placed in the sample cell, which allowed the disk to move vertically along the cell tube. Prior to the FT-IR measurements, the disk was treated under a dynamic vacuum ($10^{-4}$ Torr) at 473 K for 2 h. After cooling the disk to room temperature, $CO_2$ was introduced into the cell via the septum with a syringe. Photoluminescence excitation spectra was recorded on a FL/FS920 spectrofluorimeter (Edinburgh Instruments) fluorescence spectrometer at room temperature; the excitation wavelength is 375 nm.

**$CO_2$ photoreduction apparatus and reactions**. The spatially separated Z-scheme system includes a $CO_2$-reduction chamber and an $O_2$-generation chamber that are divided by a Nafion membrane (The circular Nafion membrane was Fe-ion exchanged before used in the reactor.) (Supplementary Fig. 6). The cubage of each reaction chamber is 300 mL. The aqueous solution containing 50 mg photocatalyst (SiC, Pt/SiC, $Cu_2O$–Pt/SiC, Pt/SiC/$IrO_x$ or $Cu_2O$–Pt/SiC/$IrO_x$) and 2 mM $FeCl_2$ and another aqueous solution containing 100 mg Pt/$WO_3$ and 2 mM $FeCl_3$ were added to the two reaction chambers, respectively. The pH of solutions was adjusted to 2.3 by adding hydrochloric acid to prevent hydrolysis of the iron ions. Prior to irradiation, ultra-pure Ar (99.9995 v%) gas was bubbled through the solution to purge any dissolved air in the $O_2$-generation compartment and filled to atmospheric pressure, while ultra-pure $CO_2$ gas was bubbled through the solution in the $CO_2$-reduction chamber and filled to atmospheric pressure. During the photoreaction, the solution in each chamber was stirred and irradiated with a 300 W xenon (Xe) lamp. The lamp with an optical filter ($\lambda \geq 420$ nm) was vertically placed at equal distance from each chamber, which makes both solutions to receive the same amount of visible light intensity. The solution in the $CO_2$-reduction chamber was sampled every 2 h and analysed by ion chromatography (IC, Thermofisher ICS 1100) after filtering catalyst. The gaseous products in $O_2$ evolution chamber were sampled every 2 h by an off-line sampling syringe (0.5 mL) and then analysed by the gas chromatography (GC, Agilent 7890B, TCD detector) using ultra-pure Ar as the carrier gas.

**Photoelectrochemical measurements**. Photoelectrochemical measurements were carried out with a BAS Epsilon workstation using a standard three-electrode electrochemical cell with a working electrode, a platinum foil as the counter electrode, and a saturated Ag/AgCl electrode as the reference. A sodium sulfate solution (0.2 M) was used as the electrolyte, and a 300 W Xe lamp ($\lambda = 320$–780 nm) as the light source. The working electrode was prepared by FTO glass pieces, which was cleaned by sonication in cleanout fluid, acetone and ethanol in sequence prior to use. The photocatalyst was dispersed in ethanol under sonication to form a suspension. A photocatalyst film was fabricated by spreading the suspension onto the conductive surface of the FTO glass.

## Data availability

The data that support the findings of this study are available from the corresponding author upon reasonable request. Source data are provided with this paper.

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

## Acknowledgements

The work is financially supported by the National Natural Science Foundation of China (grants no. 21673042, 51702053, 21673043, 21972020).

## Author contributions

Y.W., Z.Z. and X.W. conceived the research. Y.W. and X.S. prepared photocatalysts and conducted all the experiments. J.L. performed the electrochemistry measurement. J.S. offered help to analyse the characterization experiment data. Y.W., Z.Z., X.W. and C.L. wrote and revised the manuscript. D.W., J.C.S.W. and X.F. gave suggestions on the experiment and writing.

## Competing interests

The authors declare no competing interests.
