## [Peer Review File · Nature Communications]

Reviewers' Comments:

Reviewer #1:

Remarks to the Author:

In this manuscript, the authors fabricated Cu₂O-Pt/SiC/IrO_x composite as an effective photocatalyst to achieve CO₂-to-HCOOH conversion and stoichiometric O₂ evolution in a spatially separated setup. The photocatalyst can be prepared by photo-deposition and electron transfer proceeds effectively following a Z-scheme configuration. I think some key aspects of the photocatalytic process should be clarified before publication of this manuscript, and my concerns are as follows:

1. During photocatalytic experiments, HCOOH is produced as the predominant CO₂ reduction product with negligible CH₄, HCHO and CO being generated. An explanation for this high selectivity is recommended.
2. In this work, Fe²⁺ is added to the CO₂RR counterpart of the setup and Fe³⁺ is added to the OER counterpart that is separated from the other with a Nafion membrane. Can Fe²⁺/Fe³⁺ permeate through the membrane? If not, after reaction for a certain period, Fe²⁺ should be depleted in CO₂RR part while simultaneously, Fe³⁺ in the OER chamber is quantitatively converted into Fe²⁺. In this case, I think the authors should examine the color change of each counterpart (caused by Fe³⁺/Fe²⁺ conversion) and check whether the reaction slows down or ceases after complete Fe³⁺/Fe²⁺ inter-conversion (for instance, the catalytic behavior beyond 8 hours without freshly added Fe³⁺/Fe²⁺).
3. In a strong reducing environment, is Cu₂O stable? Similarly, is Fe³⁺ or Fe²⁺ stable in a water-oxidation system? Relative evidence should be given, such XPS and STEM-EDS.
4. We noticed that the total number of e⁻ in the typical CO₂ reduction for 8h was about 0.7 mmol, and the number of e⁻ provided by Fe²⁺ solution was about 0.6 mmol. We suggest that the reaction should be operated more than 16h to ensure the roles of Fe³⁺/Fe²⁺.
5. If Nafion membrane can relay the redox couple Fe²⁺/Fe³⁺ freely, why in the beginning of the reaction the Fe²⁺ and Fe³⁺ were separately located in the CO₂-reduction compartment and the H₂O-oxidation compartment? What would it be when Fe²⁺/Fe³⁺ are evenly dispersed in both cell?
6. The low performance of one-pot reaction may not from the coexistence of HCOOH and O₂ but the competing reactions from the reduction of Fe³⁺ and oxidation of Fe²⁺ with high concentration of Fe³⁺ and Fe²⁺.
7. It seemed that two different chambers were contacted with Fe³⁺/Fe²⁺, but the reaction rate was independent of concentrations of redox couple (Figure 6a). We wonder why the ratio of product was stoichiometric in two independence chamber. Does the Pt capacity of Pt/WO₃ influence this process? Relative experiment data should be given.
8. As demonstrated in Figure 9a, Pt acts as a mediator between SiC and Cu₂O. In Figure 6c, control experiments with Cu₂O/SiC/IrO_x in which Pt is absent and Pt-Cu₂O/SiC/IrO_x where Pt and Cu₂O are deposited in a reversed order are recommended, in order to reinforce the function of Pt in this work.
9. Line 130, the authors said that in Figure 2c and S4, the distributions of Pt and Cu in the selected area are coincident and appear at almost the same position. While the corresponding EDS mapping profiles for Pt-L and Cu-K don't look exactly the same, especially in Figure S4. Can more direct evidence be provided to prove the existence of core-shell structure between them, rather than the deposition of Cu₂O on the whole material surface?
10. The setup photographed in Figure S6, in many ways, resembles the H-cell in electrocatalysis. What is the applicability of the Cu₂O-Pt/SiC/IrO_x composite material for electrocatalysis or photo-electrocatalysis?
11. In this work, especially in Table S2, the catalytic performances of catalysts are evaluated by the rate of HCOOH generation. What about the solar-to-HCOOH conversion efficiency for Cu₂O-Pt/SiC/IrO_x under optimum conditions?
12. According to the SI, both the ¹³CO₂ and D₂O were employed as the reactants in the isotope labelling comparison reaction instead of CO₂ and H₂O, respectively. While the authors only provided the results of ¹³C in Line 206. What was the result of oxygen labeling?

13. In the table shown in Figure 8, considering that all lifetimes were quite short, was any effort expended to ensure that the temporal response was independent of optical pulse energy?
14. In Figure 6b, cycle experiment of HCOOH evolution in the spatially separated reaction system was measured. Since the system can operate without sacrificial reagents, why not measure the stability of the system for a long time instead of cycle experiment? Whether the generation of oxygen will affect the relay of Fe²⁺/Fe³⁺?
15. Line 318, the authors said that the indirect Z-scheme electron transfers from the CB of SiC to the valent band (VB) of Cu₂O would be accelerated by the Pt nanoparticles located between Cu₂O and SiC, due to the excellent conductivity and high work function of Pt. So, may be the catalytic efficiency of Cu₂O/SiC and Cu₂O/Pt/SiC should be compared to illustrate the role of Pt.

Reviewer #2:

Remarks to the Author:

This work described the direct and indirect Z-Scheme system composed by Cu₂O-Pt/SiC/IrO_x used in artificial photosynthesis to produce HCOOH and O₂ simultaneously with high efficiency. Interestingly, the separated reaction chambers connected with Nafion membrane were adopted and the artificial photosynthesis was realized with high efficiency. As a result, under visible light irradiation, the production rate of HCOOH and O₂ is up to 896.7 and 440.7 μmol g⁻¹ h⁻¹ respectively, which is much higher than that of the reported materials. To make this work more scientific and rigorous, major revision is needed before publishing in Nature Communications. My comments and suggestions are the following:

1. As mentioned in this work, the core-shell like structure composed of Cu₂O and Pt on the SiC surface was constructed. The author claimed that the Cu₂O was deposited on Pt. Have the authors overruled the growth of Cu₂O particles on the Pt surface over time? And is there any evidence to describe the interaction between Cu₂O and Pt? According to Figure 2a, Cu₂O was just surrounded with Pt and most of Cu₂O was located on SiC surface. It was unavoidable that Cu₂O was formed on the SiC with the reduction process because the excited electrons also can be existed on SiC surface. Thus, I am afraid that the illustration of the location of Cu₂O in the target system in Figure 1 is not accurate.
2. What is the special advantage of the indirect Z-Scheme system? Without the presence of Pt, the electrons still can transfer from SiC to Cu₂O along with the energy diagram in Figure 9. To confirm the merit of the indirect Z-Scheme, the control experiment with two direct Z-Scheme (Cu₂O/SiC/IrO_x) should be supplemented.
3. In this work, two reaction chambers were adopted and this separated half reaction system was effective in the production of the target product. From the results in Figure 6, this device was more useful than that of Cu₂O-Pt/SiC/IrO_x system in enhancement of the artificial photosynthesis efficiency. But why both the proton reduction in Pt/WO₃ system and the oxidation of water in Cu₂O-Pt/SiC/IrO_x system can be suppressed with the presence of redox couple Fe²⁺/Fe³⁺? What is the driving force of the diffusion of Fe²⁺/Fe³⁺ between in the two system?
4. It is impossible all the Pt is in the form of metallic state, especially in the Cu₂O-Pt/SiC and Cu₂O-Pt/SiC/IrO_x. Because the XPS peaks are wide and the area ratio of Pt Peaks is not 4:3. Thus XPS peaks of Pt, Cu₂O and Ir should be reanalyzed.
5. Why the selectivity of the HCOOH is high? The possible HCOOH production process and the intermediate products should be given with the help of theoretical calculation or testing technology.

6. Please give the AQY of this artificial photosynthesis.

7. Please check the descriptions carefully. In page 11, the "Pt/TiO₂" in line 279 and 284 should be corrected. Keep the identical form in figures in the whole manuscript, especially the sequence.

Reviewer #3:

Remarks to the Author:

This manuscript described an interesting spatial separation photocatalytic system for CO₂ reduction to HCOOH. Remarkable activity for COOH was achieved in this work, while stoichiometric O₂ was produced. Still, it needs a minor revision before we accepted this manuscript:

1. Author described that "no substantial change after the deposition of various cocatalysts (Figure S2)" when discussing the influence of surface modification to the BET specific surface area of SiC. However, according to Figure S2, it cannot be neglected that the BET specific surface of Cu₂O-Pt/SiC/IrO_x (~14 m² g⁻¹) reduced about 22 % compared to that of bare SiC (~18 m² g⁻¹). More accurate description should be added.

2. When discussing the distribution of IrO_x on the surface of SiC, it was described that "it is separated from Cu₂O-Pt on the SiC surface". In Figure 2, Ir is evenly distributed on the surface of SiC, even in the blank background of EDS. Despite the overlap of Cu and Pt on the EDS, Cu is also evenly distributed on SiC. This makes the conclusion of spatial separation of Cu and Ir lack of support. The EDS profile for Ir may also lead to a conclusion that Ir is not firmly combined with SiC. More solid evidence to your conclusion should be given.

3. Some references on CO₂ photoreduction should be cited, such as Nature Communications 2018, 9, 1252 etc.

Point-by-point Response to the reviewer 1#

Reviewer #1 (Remarks to the Author):

In this manuscript, the authors fabricated $\text{Cu}_2\text{O-Pt/SiC/IrO}_x$ composite as an effective photocatalyst to achieve CO_2 -to- HCOOH conversion and stoichiometric O_2 evolution in a spatially separated setup. The photocatalyst can be prepared by photo-deposition and electron transfer proceeds effectively following a Z-scheme configuration. I think some key aspects of the photocatalytic process should be clarified before publication of this manuscript, and my concerns are as follows:

Question 1. During photocatalytic experiments, HCOOH is produced as the predominant CO_2 reduction product with negligible CH_4 , HCHO and CO being generated. An explanation for this high selectivity is recommended.

Response 1. *We really appreciate your suggestion. For photocatalytic CO_2 reduction, the possible products include CO , HCOOH , HCHO , CH_3OH , and CH_4 compounds. High selectivity to product HCOOH rather than other products is necessary to discuss in this work. Because the product distribution has been documented to depend on many factors, such as the photocatalyst states (defects, crystal faces, doping...), reaction conditions (temperature, pH, CO_2 concentration, reactant phase...), co-catalyst and so on, the detailed mechanism of the reaction is far from understood, no mention to control the selectivity [ACS Catal., 2016, 6, 2018-2025; Applied Surface Science, 2019, 483, 363-372]. It is generally recognized that the photocatalytic reduction of CO_2 follows two alternative pathways dependently on the reaction conditions, (i) formaldehyde pathway: $\text{CO}_2 \rightarrow \text{HCOOH} \rightarrow \text{H}_2\text{CO} \rightarrow \text{CH}_3\text{OH} \rightarrow \text{CH}_4$, or (ii) carbene pathway: $\text{CO}_2 \rightarrow \text{CO} \rightarrow \text{C} \rightarrow \text{CH}_3 \rightarrow \text{CH}_3\text{OH/CH}_4$. CO_2 molecules with monodentate binding to the catalyst surface are generally favorable to the formation of the carboxyl (hydroxyformyl) radical $\cdot\text{COOH}$ and then recombines with a hydrogen radical H to form formic acid [Angew. Chem. Int. Ed., 2013, 52, 7372-7408]. This is the favored reaction in a medium with a high dielectric constant such as water, especially in acidic aqueous solution for supplying more amounts of protons [Chem. Sci., 2015, 6, 7213-7221]. There are two possible reasons that lead to high selectivity of HCOOH on our photocatalysts. (i) For SiC-based catalyst system, in our previous work, it has been shown that the MoS_2/SiC photocatalyzed the CO_2 reduction into CH_4 undergoing HCOOH , HCHO , and CH_3OH intermediates on SiC surface by the hydrogenation pathways under the atmosphere of H_2O vapour [J. Am. Chem. Soc., 2018, 140, 14595-14598]. (ii) Our photocatalytic experiment was performed in the acidic aqueous solution with $\text{pH} = 2.3$ and the reduction of CO_2 occurred on Cu_2O surface, which are favorable to both formation of the carboxyl (hydroxyformyl) radical intermediate $\cdot\text{COOH}$ and formic acid by recombination of the monodentate adsorption CO_2 with a hydrogen ion on the catalyst surface. In the revised manuscript, we have added the following discussion of the reason for high selectivity. "High selectivity of product HCOOH may be related with the different reaction mechanism dependent on the reaction conditions, such as the photocatalyst states (defects, crystal faces, doping...), reaction conditions (temperature, pH, CO_2 concentration, reactant phase...), co-catalyst and so on. In gas (CO_2 , H_2O vapour)-solid (catalyst.) mode, it has been found that the SiC-based composite (MoS_2/SiC) photocatalyzed the CO_2 reduction*

into CH₄ undergoing HCOOH, HCHO, and CH₃OH intermediates on SiC surface by the hydrogenation pathways¹. In the present work, the reaction is conducted in gas (CO₂)-liquid (H₂O)-solid(catalyst) mode and the acidic aqueous solution. In addition, the reduction of CO₂ occurs on the Cu₂O sites rather than SiC, which is helpful for formation of the carboxyl (hydroxyformyl) radical intermediate •COOH and HCOOH^{2,3}. ”

1. *J. Am. Chem. Soc.*, 2018, 140, 14595-14598.

2. *Angew. Chem. Int. Ed.*, 2013, 52, 7372-7408.

3. *Chem. Sci.*, 2015, 6, 7213-7221.

Question 2. In this work, Fe²⁺ is added to the CO₂RR counterpart of the setup and Fe³⁺ is added to the OER counterpart that is separated from the other with a Nafion membrane. Can Fe²⁺/Fe³⁺ permeate through the membrane? If not, after reaction for a certain period, Fe²⁺ should be depleted in CO₂RR part while simultaneously, Fe³⁺ in the OER chamber is quantitatively converted into Fe²⁺. In this case, I think the authors should examine the color change of each counterpart (caused by Fe³⁺/Fe²⁺ conversion) and check whether the reaction slows down or ceases after complete Fe³⁺/Fe²⁺ inter-conversion (for instance, the catalytic behavior beyond 8 hours without freshly added Fe³⁺/Fe²⁺).

Response 2. *Thanks for very professional review. Both the reaction cell and the Nafion membrane were supplied by our co-author Prof. Jeffrey C. S. Wu. The modified Nafion membrane for the phenomenon of Fe²⁺/Fe³⁺ permeation in the spatially-separated Z-scheme system had been studied previously in detail [J. Membrane Sci., 2011, 382, 291-299]. Either Fe³⁺ or Fe²⁺ ions are able to permeate through the membrane. The counter diffusion Fe²⁺/Fe³⁺ occurs by concentration gradient during the photoreaction. The diffusion rate of Fe²⁺/Fe³⁺ eventually matched the rate of redox reactions in two compartments. In fact, we observed the color of Fe³⁺ gradually changed during the reaction from one side to another side, and eventually color became stable, indicating the balance of Fe²⁺/Fe³⁺ in both sides when redox reactions reach steady state. As prolonging the CO₂ reduction reaction time to 16 h, the HCOOH evolution was gradually slow down for more than 8 h (Figure S8). However, it should be noted that the pH value of solution was also increased from 2.3 of the initial solution to 4.5 after 12 h reaction due to the consumption of H⁺ in the process of CO₂ reduction. The increased pH leads to the formation of FeOOH or Fe(OH)₃ in the reaction system. The sample after the reaction was characterized by XPS spectra. Fe 2p_{5/2} peak with a binding energy at 713.6 eV corresponding to FeOOH was observed on the Cu₂O-Pt/SiC/IrO_x surface (Figure S9). The change in the solution pH value and FeOOH deposition maybe result in the decrease of photocatalytic CO₂ reduction. Therefore, to keep the stability of the CO₂ reduction into HCOOH, the Cu₂O-Pt/SiC/IrO_x was centrifugated out and put into the fresh Fe²⁺ solution at each 8 h reaction.*

Figure S8. The HCOOH evolution with the reaction time prolonging to 16 h

Figure S9. The Fe 2p XPS spectra of the Cu₂O-Pt/SiC/IrO_x sample after 16 h reaction

Question 3. In a strong reducing environment, is Cu₂O stable? Similarly, is Fe³⁺ or Fe²⁺ stable in a water-oxidation system? Relative evidence should be given, such XPS and STEM-EDS.

Response 3. Thanks for your important suggestions. The Cu₂O-Pt/SiC/IrO_x sample after the reaction was characterized by XPS spectra (Figure S10). It shows that no notable change in the binding energy of Pt, Ir and Cu species is observed for the Cu₂O-Pt/SiC/IrO_x samples before and after the reaction. This indicates that the chemical states of Pt, Ir and Cu species are almost not altered after the reaction. Moreover, the relative intensity of Cu in the XPS spectra of the Cu₂O-Pt/SiC/IrO_x sample after the reaction is roughly consistent with the one before reaction. These indicate that the Cu₂O remain basically stable under the reaction conditions.

In theory, the O₂ can slowly induce the oxidation of Fe²⁺ into Fe³⁺, which can be a counter reaction affecting the efficiency. Nevertheless, the evolved O₂ from water oxidation is favorably escaped from the solution into the gas phase under stirring. Moreover, only about 0.175 mmol of O₂ (3.9 mL) was released during 8 h reaction, which is much lower than that of Fe²⁺. Both results can minimize the possibility of Fe²⁺ oxidation by the evolved O₂.

Figure S10. The XPS spectra of the $\text{Cu}_2\text{O-Pt/SiC/IrO}_x$ sample before and after cycle reaction

Question 4. We noticed that the total number of e^- in the typical CO_2 reduction for 8 h was about 0.7 mmol, and the number of e^- provided by Fe^{2+} solution was about 0.6 mmol. We suggest that the reaction should be operated more than 16 h to ensure the roles of $\text{Fe}^{3+}/\text{Fe}^{2+}$.

Response 4. *Thanks for your suggestions. As mentioned above, both Fe^{3+} and Fe^{2+} ions can be diffused through Nafion membrane due to the concentration difference between two sides of Nafion membrane. $\text{Fe}^{3+}/\text{Fe}^{2+}$ redox couple makes the reduction of CO_2 to HCOOH on $\text{Cu}_2\text{O-Pt/SiC/IrO}_x$ and the oxidation of H_2O to O_2 on Pt/WO_3 integrated into the one system like the natural photosynthetic systems. We have performed the controlling reduction half-reaction only using $\text{Cu}_2\text{O-Pt/SiC/IrO}_x$ in the Fe^{2+} solution. The HCOOH yield is about $484.0 \mu\text{mol g}^{-1} \text{h}^{-1}$, which is much lower than that in the spatially separated reaction system. This indicates the important role of $\text{Fe}^{3+}/\text{Fe}^{2+}$ redox couple as the electron-transfer mediator in the spatially separated reaction system. We have supplemented the CO_2 reduction reaction for more than 8 h. As prolonging the reaction time to 16 h, the CO_2 reduction reaction is gradually slow down for more than 8 h (Figure S8). However, it should be noted that the pH value of solution was also increased from 2.3 of the initial solution to 4.5 after 12 h reaction due to the consumption of H^+ in the process of CO_2 reduction. The increased pH leads to the appearance of FeOOH or $\text{Fe}(\text{OH})_3$ precipitation in the reaction system. The sample after the reaction was characterized by XPS spectra (Figure S9). $\text{Fe } 2p_{3/2}$ peak with a binding energy at 713.6 eV can be observed, which is belonged to the FeOOH on the $\text{Cu}_2\text{O-Pt/SiC/IrO}_x$ surface. The change in the solution pH value and FeOOH deposition maybe result in the decrease of photocatalytic CO_2 reduction.*

Question 5. If Nafion membrane can relay the redox couple $\text{Fe}^{2+}/\text{Fe}^{3+}$ freely, why in the beginning of the reaction the Fe^{2+} and Fe^{3+} were separately located in the CO_2 -reduction compartment and the H_2O -oxidation compartment? What would it be when $\text{Fe}^{2+}/\text{Fe}^{3+}$ are evenly dispersed in both cell?

Response 5. *Thanks for your questions. When we dispersed $\text{Fe}^{2+}/\text{Fe}^{3+}$ mixed solution in both cell of the separated reaction system, the HCOOH products was generated stably with the reaction time, as shown in Figure*

S23. The evolution rate of HCOOH in the Fe^{2+}/Fe^{3+} mixed solution ($\sim 629.8 \mu\text{mol g}^{-1} \text{h}^{-1}$) is slightly lower than that for the Fe^{2+} and Fe^{3+} separately located in the CO_2 -reduction compartment and the H_2O -oxidation compartment ($\sim 896.6 \mu\text{mol g}^{-1} \text{h}^{-1}$). Obviously, the separate location of Fe^{2+} and Fe^{3+} in the CO_2 -reduction compartment and the H_2O -oxidation compartment in the beginning of the reaction is favorable to the highly efficient reaction in the separated reaction system. The separate use of $FeCl_2$ in the CO_2 -reduction compartment and $FeCl_3$ in the H_2O -oxidation compartment, respectively, can improve photoreaction at the beginning [Int. J. Hydrogen Energ., 2010, 35, 1523-1529]. When the redox reaction reached steady state, the concentrations of Fe^{2+}/Fe^{3+} also reach a constant gradient in both sides to maintain counter diffusion. The Fe^{2+} and Fe^{3+} are not evenly dispersed in either cell according to previous study [Int. J. Hydrogen Energ., 2010, 35, 1523-1529] due to osmosis of Fe ions both sides. Even though the concentration of Fe^{2+} and Fe^{3+} is equal in the reaction compartment, the H_2O -oxidation and CO_2 -reduction can proceed over the photocatalysts due to the different adsorption property of Fe^{2+}/Fe^{3+} on photocatalyst surface. For example, Fe^{3+} is more favorable to adsorb on WO_3 surface than Fe^{2+} [Chem. Lett., 2017, 46, 221-224]. This makes WO_3 sustainably produce O_2 from H_2O -oxidation in the mixed Fe^{2+}/Fe^{3+} solution combining with reduction photocatalyst in Z-scheme reactions.

Figure S23. The HCOOH evolution in the separated reaction system with Fe^{2+}/Fe^{3+} even dispersion in both cell

Question 6. The low performance of one-pot reaction may not from the coexistence of HCOOH and O_2 but the competing reactions from the reduction of Fe^{3+} and oxidation of Fe^{2+} with high concentration of Fe^{3+} and Fe^{2+} .

Response 6. Thanks for your questions. In the one-pot reaction system, the backward reaction of HCOOH re-oxidization by O_2 should be one of the reasons for low evolution of HCOOH and O_2 . At the same time, both Fe^{3+} competition with CO_2 for the generated electron and Fe^{2+} competing with H_2O for the photogenerated holes as backward reactions also occur. But effects of the later could be weaker than that of the former, because if no adding Fe^{3+} and Fe^{2+} , the evolution of HCOOH and O_2 is lower due to no cooperation of CO_2 reduction and H_2O

oxidation on separated photocatalysts (see the first column in Table S4). We have revised the incorrect understanding in the revised manuscript.

Table S4 HCOOH evolution in the one-pot reaction system

Samples	one-pot reactor (No Fe^{2+} & Fe^{3+}) ($\mu\text{mol g}^{-1} \text{h}^{-1}$)	one-pot reactor (Fe^{2+} & Fe^{3+}) ($\mu\text{mol g}^{-1} \text{h}^{-1}$)	separated reactor (Fe^{2+} & Fe^{3+}) ($\mu\text{mol g}^{-1} \text{h}^{-1}$)
SiC	0.63	1.70	24.06
Pt/SiC	1.37	3.55	57.74
$\text{Cu}_2\text{O-Pt/SiC}$	6.83	18.65	304.65
Pt/SiC/ IrO_x	11.09	30.39	472
$\text{Cu}_2\text{O-Pt/SiC/IrO}_x$	22.71	61.54	896.71

Question 7. It seemed that two different chambers were contacted with $\text{Fe}^{3+}/\text{Fe}^{2+}$, but the reaction rate was independent of concentrations of redox couple (Figure 6a). We wonder why the ratio of product was stoichiometric in two independence chambers. Does the Pt capacity of Pt/ WO_3 influence this process? Relative experiment data should be given.

Response 7. Thanks for your questions. The reduction of CO_2 to HCOOH and the oxidation of H_2O to O_2 are seemingly independent each other, but they are closely related, because the counter permeation rates of $\text{Fe}^{3+}/\text{Fe}^{2+}$ through membrane are much higher (near 10 times) than the rates of redox reaction according to previous study [Int. J. Hydrogen Energ., 2010, 35, 1523-1529]. So it looks like the reaction rate was independent of concentrations of redox couple because the rate-limiting step is photoreaction. If the reduction reaction is faster than the oxidation reaction (i.e. HCOOH yield > $1/2\text{O}_2$ yield) and independent, then Fe^{2+} concentration in the reduction chamber would decrease with reaction until zero. Inversely, Fe^{3+} concentration would decrease gradually in the oxidation reaction chamber. In the case, the both reaction rates would decrease with increasing reaction time. We think the ratio of product may not be in stoichiometric one in two independence chambers at the beginning of the reaction, but when the reaction reaches to its steady state, the ratio of product is surely stoichiometric in two independence chambers due to the compensation effect between both reactions under the mediation of $\text{Fe}^{3+}/\text{Fe}^{2+}$ pairs.

According to the suggestion, we had supplemented 0.5, 1.0 and 1.5 wt% of Pt in Pt/ WO_3 to investigate the influence of Pt contents on the photocatalytic activity in separated reaction system. As shown in the Figure S7, decreasing the Pt contents to 0.5 wt% of Pt/ WO_3 , both the O_2 and HCOOH evolutions are decreased to 296.6 and 618.7 $\mu\text{mol g}^{-1} \text{h}^{-1}$, respectively. However, the ratio of product O_2 to product HCOOH is close to the stoichiometric one of reaction ($2\text{CO}_2 + 2\text{H}_2\text{O} \rightarrow 2\text{HCOOH} + \text{O}_2$) in the whole reaction process. When the Pt

content is increased to 1.5 wt% of Pt/WO₃, the O₂ and HCOOH evolutions are almost the same with the case of using 1.0 wt% Pt contents of Pt/WO₃. The 1.0 wt% Pt content is optimal for the Pt/WO₃ in the separated reaction system. Obviously, the Pt contents of Pt/WO₃ can influence the evolution rate of O₂ and HCOOH, but the stoichiometric ratio of O₂ and HCOOH product is independent on the Pt contents. This is because Fe³⁺/Fe²⁺ redox couple acts as the electron mediator to link CO₂ reduction and H₂O oxidation reactions, which can keep the balance of photogenerated holes and electrons for the stoichiometric reaction (2CO₂ + 2H₂O → 2HCOOH + O₂) in the whole reaction process.

Figure S7. The effect of Pt contents of Pt/WO₃ on the O₂ and HCOOH evolution in the separated reaction system.

Question 8. As demonstrated in Figure 9a, Pt acts as a mediator between SiC and Cu₂O. In Figure 6c, control experiments with Cu₂O/SiC/IrO_x in which Pt is absent and Pt-Cu₂O/SiC/IrO_x where Pt and Cu₂O are deposited in a reversed order are recommended, in order to reinforce the function of Pt in this work.

Response 8. It is generally accepted that the metal conductor in the indirect Z-Scheme systems shows the stronger ability for charge transfer than in the case of the solid-solid contact interface in the direct Z-Scheme systems due to the difference in their electrical resistances [Adv. Mater., 2014, 26, 4920-4935]. Therefore, the Pt embedded at the interface between SiC and Cu₂O to form the indirect Z-Scheme structure should be more efficient for the electron transfer than the direct Z-Scheme of Cu₂O/SiC. The additional experiment using Cu₂O/SiC/IrO_x or Pt-Cu₂O/SiC/IrO_x as photocatalyst for the CO₂ reduction reaction was performed to compare with Cu₂O-Pt/SiC/IrO_x (Figure S21). The results show that Cu₂O/SiC/IrO_x or Pt-Cu₂O/SiC/IrO_x has a much lower activity of CO₂ reduction into HCOOH than Cu₂O-Pt/SiC/IrO_x. This confirms the merit of the indirect Z-Scheme for the Cu₂O-Pt/SiC/IrO_x. The discussion is added in the revised manuscript.

Figure S21. The comparison of HCOOH evolution over the various photocatalysts.

Question 9. Line 130, the authors said that in Figure 2c and S4, the distributions of Pt and Cu in the selected area are coincident and appear at almost the same position. While the corresponding EDS mapping profiles for Pt-L and Cu-K don't look exactly the same, especially in Figure S4. Can more direct evidence be provided to prove the existence of core-shell structure between them, rather than the deposition of Cu₂O on the whole material surface?

Response 9. We appreciate your scientific comments. We carefully check the EDS mapping of Cu element distribution, and take the pictures of STEM-EDS mapping again. Besides the Cu₂O deposition on Pt, a part of Cu₂O is located on SiC surface. However, the bright spot of Pt element in EDS mapping is coincident with that of Cu element distribution. Moreover, the Pt particles covered by Cu₂O on SiC surface can also be verified by the low energy ion scattering (LEIS) spectra (Figure 3a and 3b). For both 3 keV ⁴He⁺ and 5 keV ²⁰Ne⁺ spectra, the peak of Pt can be observed in Pt/SiC (line ii), while the Pt peak disappeared when Cu₂O was loaded on Pt/SiC (line iii). Meanwhile, the peak belongs to Cu appears in Cu₂O-Pt/SiC (line iii), which indicates that the Pt is covered by Cu₂O to form a core-shell structure. The new result has already added in the revised manuscript (Figure 2).

Figure 2 (a) TEM, (b) HRTEM images and (c) STEM image and corresponding EDS mapping profiles for C-K, Si-K, Pt-L, Cu-K and Ir-L of $\text{Cu}_2\text{O-Pt/SiC/IrO}_x$.

Figure 3 HS-LEIS spectra using (a) 3 keV $^4\text{He}^+$ and (b) 5 keV $^{20}\text{Ne}^+$ for the samples: (i) SiC, (ii) Pt/SiC, (iii) $\text{Cu}_2\text{O-Pt/SiC}$, (iv) Pt/SiC/IrO_x , and (v) $\text{Cu}_2\text{O-Pt/SiC/IrO}_x$.

Question 10. The setup photographed in Figure S6, in many ways, resembles the H-cell in electrocatalysis. What is the applicability of the $\text{Cu}_2\text{O-Pt/SiC/IrO}_x$ composite material for electrocatalysis or photo-electrocatalysis?

Response 10: Thank you for the kind suggestion. The used reactor cell in our work is like in principle as the H-cell in electrocatalysis with the separated oxidation and reduction compartments, but the used cell lacks of electrode mounting position, and the air tightness cannot be guaranteed if we use it for electrocatalysis or photo-electrocatalysis. It is difficult to perform the electrocatalysis or photo-electrocatalysis with the current reaction cell. We would like to explore a good way to apply the catalyst to electrocatalysis or photo-electrocatalysis in future research.

Question 11. In this work, especially in Table S2, the catalytic performances of catalysts are evaluated by the rate of HCOOH generation. What about the solar-to-HCOOH conversion efficiency for $\text{Cu}_2\text{O-Pt/SiC/IrO}_x$ under optimum conditions?

Response 11. Thanks for your suggestions. We have performed the wavelength dependent activity of HCOOH evolution, and thus the apparent quantum yield (AQY) was calculated, as shown in Figure S17. Obviously, the AQY of HCOOH evolution on the optimal $\text{Cu}_2\text{O-Pt/SiC/IrO}_x$ sample is well matched with the optical absorption spectra of SiC. With 400 nm light irradiation, the AQY of HCOOH evolution can be reached about 1.44%. The above result was added in the revised manuscript.

Figure S17. The apparent quantum yield of HCOOH evolution with the wavelength of irradiation light for the separated reaction system.

Question 12. According to the SI, both the $^{13}\text{CO}_2$ and D_2O were employed as the reactants in the isotope labelling comparison reaction instead of CO_2 and H_2O , respectively. While the authors only provided the results of ^{13}C in Line 206. What was the result of oxygen labeling?

Response 12. We are sorry for the mistake. We have intended to use H_2^{18}O to label the O_2 coming from the H_2O oxidation over Pt/WO_3 photocatalyst other than O_2 contamination. However, the H_2^{18}O reagent is very expensive (1 mL H_2^{18}O required 1400 RMB), and the real reaction solution has a volume of 300 mL. Thus, the high cost makes it impossible to perform the oxygen labeling experimental with 300 mL H_2^{18}O . The Pt/WO_3 is a well-known photocatalyst for the water oxidation to produce O_2 in the presence of electron acceptor sacrificial agent such as Ag^+ , Fe^{3+} et al. Many studies have reported the O_2 evolution from water oxidation over Pt/WO_3 in the presence of Fe^{3+} [J. Photochem. and Photobiology A: Chemistry, 1999, 122, 175-183; Int. J. Hydrogen Energ., 2010, 35, 1523-1529]. To confirm the O_2 evolution, we used 2 mL H_2^{18}O with Pt/WO_3 in the presence of Fe^{3+} to simulate the reaction under visible light irradiation, and the products were analyzed with Mass Spectra. The result shows that the main peak at $m/z = 36$ corresponding to $^{18}\text{O}_2$ is observed (Figure S12). This confirms the produced O_2 resulting from H_2O oxidation. This result was added in the revised manuscript.

Figure S12. The mass spectra of product O_2 over Pt/WO_3 in the 2 ml H_2^{18}O solution

Question 13. In the table shown in Figure 8, considering that all lifetimes were quite short, was any effort expended to ensure that the temporal response was independent of optical pulse energy?

Response 13. This is indeed a question that worth considering. We planned to re-measure the fluorescence lifetime spectrum of sample. Unfortunately, however, we have not been allowed to do the experiment during preventing of the spread of the COVID-19 virus from 23 Jan 2020, and we don't know when it will go back to work. We originally used the time-resolved photoluminescence spectroscopy to confirm the transfer and separation efficiency of photogenerated charges of SiC-based catalysts. For the different SiC-based catalysts,

especially the $\text{Cu}_2\text{O-Pt/SiC/IrO}_x$, the photocurrents, AC impedance and the steady state photoluminescence (PL) has demonstrated the enhancement of photogenerated charge separation with the cocatalyst $\text{Cu}_2\text{O-Pt}$ and IrO_x modification as compared with the parent SiC. The characterizations of the time-resolved photoluminescence spectroscopy are consistent with the above results, confirming the more efficient charge separation on $\text{Cu}_2\text{O-Pt/SiC/IrO}_x$. Moreover, the lifetime τ value of the parent SiC is about 1.2 ns measured by the time-resolved photoluminescence spectroscopy in this work, which is consistent with the value of the reported literature [Appl. Phys. Lett. 2000, 76, 2550-2552]. Although the PL lifetime of the parent SiC or $\text{Cu}_2\text{O-Pt/SiC/IrO}_x$ is quite short, and the accurate measurement of PL lifetime is highly demanding.

Question 14. In Figure 6b, cycle experiment of HCOOH evolution in the spatially separated reaction system was measured. Since the system can operate without sacrificial reagents, why not measure the stability of the system for a long time instead of cycle experiment? Whether the generation of oxygen will affect the relay of $\text{Fe}^{2+}/\text{Fe}^{3+}$?

Response 14. This is a very academic question indeed. We had performed the CO_2 reduction reaction for more than 8 h. As prolonging the reaction time to 16 h, the CO_2 reduction reaction is gradually slow down (Figure S8). One of the reasons for the efficiency droop is that the reaction system was closed, the product O_2 pressure would increase with prolonging reaction time. This would suppress further evolution of O_2 in the oxidation chamber and then affect the HCOOH formation. On the other hand, in the reduction chamber, the following reactions occurred

With increasing reaction time, OH concentration would increase and leads to the reaction,

The formed FeOOH would deposit at the catalyst surface and Nafion membrane to result in the activity decline. This inference was supported by the following observations. The pH value of solution was increased from 2.3 of the initial solution to 4.5 after 12 h reaction due to the consumption of H^+ in the process of CO_2 reduction. The sample after the reaction was characterized by XPS spectra (Figure S9). Fe $2p_{5/2}$ peak with a binding energy at 713.6 eV is observed, which confirms the FeOOH on the $\text{Cu}_2\text{O-Pt/SiC/IrO}_x$ surface. However, in the spatially separated reaction system, the two effects would increase to some extent. The cycle experiment mode could decrease the two effects. In this sense, the reaction system needs the further improvement.

Question 15. Line 318, the authors said that the indirect Z-scheme electron transfers from the CB of SiC to the valent band (VB) of Cu_2O would be accelerated by the Pt nanoparticles located between Cu_2O and SiC, due to the

excellent conductivity and high work function of Pt. So, may be the catalytic efficiency of Cu₂O/SiC and Cu₂O/Pt/SiC should be compared to illustrate the role of Pt.

Response 15. Thanks for your suggestions. The Cu₂O/SiC without Pt was prepared for the control experiment of CO₂ reduction. The Cu₂O/SiC displays the HCOOH evolution of about 40.5 μmol g⁻¹ h⁻¹, little higher than that of the parent SiC but much lower than that of Cu₂O-Pt/SiC samples (Figure S21). This means that contribution of the Cu₂O deposited on SiC surface for the activity is very small, the increased activity is because of the embedding of Pt in the interface between Cu₂O and SiC. The photocatalytic CO₂ reduction performances of Cu₂O/SiC/IrO_x and Pt-Cu₂O/SiC/IrO_x were also compared with that of Cu₂O-Pt/SiC/IrO_x (Figure S21). Much lower activities of Cu₂O/SiC/IrO_x and Pt-Cu₂O/SiC/IrO_x than Cu₂O-Pt/SiC/IrO_x indicates likewise that the Pt embedded at the interface of Cu₂O and SiC to form the indirect Z-Scheme structure is beneficial to the transfer of photogenerated electrons from SiC to Cu₂O and therefore enhances the CO₂ reduction efficiency to HCOOH. This is because the metal conductor in the indirect Z-Scheme systems shows the stronger ability for charge transfer than in the case of the solid-solid contact interface in the direct Z-Scheme systems due to the difference in their electrical resistances [Adv. Mater., 2014, 26, 4920-4935]. This has been added in the revised manuscript.

Figure S21. The comparison of HCOOH evolution over the various photocatalysts.

Point-by-point Response to the reviewer 2#

Reviewer #2 (Remarks to the Author):

This work described the direct and indirect Z-Scheme system composed by Cu₂O-Pt/SiC/IrO_x used in artificial photosynthesis to produce HCOOH and O₂ simultaneously with high efficiency. Interestingly, the separated reaction chambers connected with Nafion membrane were adopted and the artificial photosynthesis was realized with high efficiency. As a result, under visible light irradiation, the production rate of HCOOH and O₂ is up to 896.7 and 440.7 μmol g⁻¹ h⁻¹ respectively, which is much higher than that of the reported materials. To make this work more scientific and rigorous, major revision is needed before publishing in Nature Communications. My comments and suggestions are the following:

Question 1. As mentioned in this work, the core-shell like structure composed of Cu₂O and Pt on the SiC surface was constructed. The author claimed that the Cu₂O was deposited on Pt. Have the authors observed the growth of Cu₂O partials on the Pt surface over time? And is there any evidence to describe the interaction between Cu₂O and Pt? According to Figure 2a, Cu₂O was just surround with Pt and most of Cu₂O was located on SiC surface. It was unavoidable that Cu₂O was formed on the SiC with the reduction process because the excited electrons also can be existed on SiC surface. Thus, I am afraid that the illustration of the location of Cu₂O in the target system in Figure 1 is not accurate.

Response 1. *We appreciate your scientific comments. We have carefully checked the EDS mapping profile of Cu element and took the pictures of STEM-EDS mapping again. Indeed, besides the Cu₂O overlapping with Pt particles, a part of Cu₂O is directly located on SiC surface. It can be seen from Figure 2 that the bright spot of Pt element in EDS mapping is coincident with that of Cu element distribution. Moreover, the Pt particles covered by Cu₂O on SiC surface can also be verified by the low energy ion scattering (LEIS) spectra (Figure 3a and 3b). For both 3 keV ⁴He⁺ and 5 keV ²⁰Ne⁺ spectra, the peak of Pt can be observed in Pt/SiC (line ii), while the Pt peak disappeared when Cu₂O was loaded on Pt/SiC (line iii). Meanwhile, the peak belonging to Cu appears in Cu₂O-Pt/SiC (line iii), which indicates that a part of Pt is covered by Cu₂O to form a core-shell structure. Since Cu₂O and IrO_x were deposited on Pt/SiC after the loading Pt and by photodeposition, deposition of Cu₂O on the Pt particles and on the surface of SiC is reasonable. The Cu₂O/SiC without Pt was prepared also for comparison. The Cu₂O/SiC displays a HCOOH evolution of about 40.5 μmol g⁻¹ h⁻¹, which is little higher than that of the parent SiC but much lower than that of Cu₂O-Pt/SiC samples (Figure S21). This confirms that contribution of the deposited Cu₂O on SiC surface for the activity is very small, but the increased activity is due to that Pt was buried beneath Cu₂O. In the revised manuscript, Figure 2 was revised and the description has changed into “In the selected area, Pt has the same distributions and appears at almost the same position as Cu. This demonstrates that the deposited Pt is fully covered with Cu₂O at the same sites on SiC surface. Nevertheless, there is also a part of Cu located on SiC surface.”*

Question 2. What is the special advantage of the indirect Z-Scheme system? Without the presence of Pt, the electrons still can transfer from SiC to Cu₂O along with the energy diagram in Figure 9. To confirm the merit of the indirect Z-Scheme, the control experiment with two direct Z-Scheme (Cu₂O/SiC/IrO_x) should be supplemented.

Response 2. It is generally accepted that the metal conductor in the indirect Z-Scheme systems shows the stronger ability for charge transfer than in the case of the solid-solid contact interface in the direct Z-Scheme systems due to the difference in their electrical resistances [Adv. Mater., 2014, 26, 4920-4935]. Therefore, the Pt embedded at the interface between SiC and Cu₂O to form the indirect Z-Scheme structure should be more efficient for the electron transfer than the direct Z-Scheme of Cu₂O/SiC. To confirm the merit of the indirect Z-Scheme, the additional photocatalysts Cu₂O/SiC/IrO_x or Pt-Cu₂O/SiC/IrO_x were also used for the CO₂ reduction reaction to compare with Cu₂O-Pt/SiC/IrO_x (Figure S21). The results show that Cu₂O/SiC/IrO_x or Pt-Cu₂O/SiC/IrO_x exerts a much lower activity of CO₂ reduction into HCOOH than Cu₂O-Pt/SiC/IrO_x. So, the Pt embedded the interface of Cu₂O and SiC to form the indirect Z-Scheme structure is beneficial to the transfer of photogenerated electrons from SiC to Cu₂O and therefore increase of the CO₂ reduction efficiency to HCOOH. In the revised manuscript, the related discussion has been changed.

Figure S21. The comparison of HCOOH evolution over the various photocatalysts.

Question 3. In this work, two reaction chambers were adopted and this separated half reaction system was effective in the production of the target product. From the results in Figure 6, this device was more useful than

that of Cu₂O-Pt/SiC/IrO_x system in enhancement of the artificial photosynthesis efficiency. But why both the proton reduction in Pt/WO₃ system and the oxidation of water in Cu₂O-Pt/SiC/IrO_x system can be suppressed with the presence of redox couple Fe²⁺/Fe³⁺? What is the driving force of the diffusion of Fe²⁺/Fe³⁺ between in the two system?

Response 3. *Pt/WO₃ is a well-known photocatalyst for the water oxidation into O₂, but the conduction band of WO₃ is located at positive potentials (approximately +0.2 to +0.3 V vs NHE). Therefore, WO₃ has not enough negative potential for the proton reduction (H⁺/H₂, 0 V vs NHE). The reduction of Fe³⁺ (Fe²⁺/Fe³⁺, +0.77 V vs NHE) can be satisfied over Pt/WO₃. In the presence of Fe³⁺ to consume the photogenerated electrons, the water oxidation into O₂ can be continuously proceed on Pt/WO₃. Although the band structure of Cu₂O-Pt/SiC/IrO_x make the oxidation of water possible, but the Fe²⁺ oxidation (Fe²⁺/Fe³⁺, +0.77 V vs NHE) is much easier than H₂O oxidation (H₂O/O₂, +1.23 V vs NHE), because the H₂O oxidation was a slow and complicated four-electron participation process, which is generally considered to be the rate-determining step of total photolysis reaction. To accelerate the CO₂ reduction over Cu₂O-Pt/SiC/IrO_x, using the Fe²⁺/Fe³⁺ redox mediator is essential to rapidly remove the photogenerated holes. The driving force of the diffusion of Fe²⁺/Fe³⁺ is the concentration gradient of Fe²⁺/Fe³⁺ in either side in the spatially-separated Z-scheme system when the photo reaction is performed.*

Question 4. It is impossible all the Pt is in the form of metallic state, especially in the Cu₂O-Pt/SiC and Cu₂O-Pt/SiC/IrO_x. Because the XPS peaks are wide and the area ratio of Pt Peaks is not 4:3. Thus XPS peaks of Pt, Cu₂O and Ir should be reanalyzed.

Response 4. *Thank you very much for your valuable comments. We have reanalyzed the XPS peaks of Pt, Cu and Ir elements, as shown in Figure 4 below. Fixing the area ratio of Pt 4f_{7/2} to Pt 4f_{5/2} to be 4:3, the Pt 4f can be fitted into three couple peaks, indicating three chemical states of Pt in the all samples. The most of Pt is in metallic state, and a slight amount of Pt²⁺ and Pt⁴⁺ species was contained in all samples. The ratio of Pt⁰ was calculated to be about 70 ± 4% of the sum Pt species for the all Pt-contained samples (Table S2). The Cu 2p peaks can be well resolved into one couple peaks, which is corresponding to the state of Cu₂O. The one doublet peaks corresponding to Ir 4f_{7/2} and Ir 4f_{5/2} is close to the state IrO_x value.*

Figure 4 (a) Pt 4f XPS spectra, (b) Cu 2p XPS spectra, and (c) Ir 4f XPS spectra of samples: (□) Pt/SiC, (□) Cu₂O-Pt/SiC, (□) Pt/SiC/IrOx, and (□) Cu₂O-Pt/SiC/IrOx.

Question 5. Why the selectivity of the HCOOH is high? The possible HCOOH production process and the intermediate products should be given with the help of theoretical calculation or testing technology.

Response 5. We really appreciate your suggestion. The similar question was mentioned by reviewer 1. For photocatalytic CO₂ reduction, the possible products include CO, HCOOH, HCHO, CH₃OH, and CH₄ compounds. High selectivity to product HCOOH rather than other products is necessary to discuss in this work. Because the product distribution has been found to depend on many factors, such as the photocatalyst states (defects, crystal faces, doping...), reaction conditions (temperature, pH, CO₂ concentration, reactant phase...), co-catalyst and so on, the detailed mechanism of the reaction is far from understood, no mention to control the selectivity [ACS Catal., 2016, 6, 2018-2025]. It is generally recognized that the photocatalytic reduction of CO₂ follows two alternative pathways dependently on the reaction conditions, (i) formaldehyde pathway: CO₂ → HCOOH → H₂CO → CH₃OH → CH₄, or (ii) carbene pathway: CO₂ → CO → C → CH₃ → CH₃OH/CH₄. CO₂ molecules with monodentate binding to the catalyst surface are generally favorable to the formation of the carboxyl (hydroxyformyl) radical ·COOH and then recombines with a hydrogen radical H· to form formic acid. [Angew. Chem. Int. Ed., 2013, 52, 7372-7408] This is the favored reaction in a medium with a high dielectric constant such as water, especially in acidic aqueous solution for supplying more amounts of protons. [Chem. Sci., 2015, 6, 7213-7221] There are two possible reasons that lead to high selectivity of HCOOH on our photocatalysts. (i) For SiC-based catalyst system, in our previous work, it has been shown that the MoS₂/SiC photocatalyzed the

CO₂ reduction into CH₄ undergoing HCOOH, HCHO, and CH₃OH intermediates on SiC surface by the hydrogenation pathways under the atmosphere of H₂O vapour [J. Am. Chem. Soc., 2018, 140, 14595-14598]. (ii) Our photocatalytic experiment was performed in the acidic aqueous solution with pH = 2.3 and the reduction of CO₂ occurred on Cu₂O surface, which are favorable to both formation of the carboxyl (hydroxyformyl) radical intermediate ·COOH and formic acid by recombination of the monodentate adsorption CO₂ with a hydrogen ion on the catalyst surface. In the revised manuscript, we have added the following discussion of the reason for high selectivity. “High selectivity of product HCOOH may be related with the different reaction mechanism dependent on the reaction conditions, such as the photocatalyst states (defects, crystal faces, doping...), reaction conditions (temperature, pH, CO₂ concentration, reactant phase...), co-catalyst and so on. In gas (CO₂, H₂O vapour)-solid (catalyst.) mode, it has been found that the SiC-based composite (MoS₂/SiC) photocatalyzed the CO₂ reduction into CH₄ undergoing HCOOH, HCHO, and CH₃OH intermediates on SiC surface by the hydrogenation pathways¹. In the present work, the reaction is conducted in gas (CO₂)-liquid (H₂O)-solid(catalyst) mode and the acidic aqueous solution. In addition, the reduction of CO₂ occurs on the Cu₂O sites rather than SiC, which is helpful for formation of the carboxyl (hydroxyformyl) radical intermediate ·COOH and HCOOH^{2,3}.”

1. J. Am. Chem. Soc., 2018, 140, 14595-14598

2. Angew. Chem. Int. Ed. 2013, 52, 7372-7408.

3. Chem. Sci., 2015, 6, 7213-7221.

Question 6. Please give the AQY of this artificial photosynthesis.

Response 6. Thanks for your suggestions. We have performed the wavelength dependent activity of HCOOH evolution, and thus the apparent quantum yield (AQY) was calculated, as shown in Figure S17. Obviously, the AQY of HCOOH evolution on the optimal Cu₂O-Pt/SiC/IrO_x sample is well matched with the optical absorption spectra of SiC. With 400 nm light irradiation, the AQY of HCOOH evolution can be reached about 1.44%. This has been added in the revised manuscript.

Figure S17. The apparent quantum yield of HCOOH evolution with the wavelength of irradiation light for the separated reaction system.

Question 7. Please check the descriptions carefully. In page 11, the “Pt/TiO₂” in line 279 and 284 should be corrected. Keep the identical form in figures in the whole manuscript, especially the sequence.

Response 7. *We are sorry for the mistake. The “Pt/TiO₂” has been corrected into “Pt/SiC”. The sequence of figures has kept the identical form in the manuscript.*

Point-by-point Response to the reviewer 3#

Reviewer #3 (Remarks to the Author):

This manuscript described an interesting spatial separation photocatalytic system for CO₂ reduction to HCOOH. Remarkable activity for COOH was achieved in this work, while stoichiometric O₂ was produced. Still, it needs a minor revision before we accepted this manuscript:

Question 1. Author described that “no substantial change after the deposition of various cocatalysts (Figure S2)” when discussing the influence of surface modification to the BET specific surface area of SiC. However, according to Figure S2, it cannot be neglected that the BET specific surface of Cu₂O-Pt/SiC/IrO_x (~14 m² g⁻¹) reduced about 22 % compared to that of bare SiC (~18 m² g⁻¹). More accurate description should be added.

Response 1. *We greatly appreciate your suggestions. Influence of cocatalyst deposition on the BET specific surface area of SiC was discussed in the revised manuscript as following: “The BET specific surface area of SiC has a slight reduction from ~18 m² g⁻¹ for bare SiC to ~14 m² g⁻¹ for Cu₂O-Pt/SiC/IrO_x (Figure S2), possibly because the cocatalysts with small particle size block the micropore structure of SiC.”*

Question 2. When discussing the distribution of IrO_x on the surface of SiC, it was described that “it is separated from Cu₂O-Pt on the SiC surface”. In Figure 2, Ir is evenly distributed on the surface of SiC, even in the blank background of EDS. Despite the overlap of Cu and Pt on the EDS, Cu is also evenly distributed on SiC. This makes the conclusion of spatial separation of Cu and Ir lack of support. The EDS profile for Ir may also lead to a conclusion that Ir is not firmly combined with SiC. More solid evidence to your conclusion should be given.

Response 2. *Because of the proximity of the Pt and Ir peaks, the low contents and the small particle sizes of Pt and IrO_x, achieving both sufficient signal and accurate spatial resolution in STEM-EDS mapping is problematic [ACS Energy Lett., 2017, 2, 244-249]. To obtain the element distribution on SiC surface as accurately as possible, we have taken the pictures of STEM-EDS mapping again by long time signal collection of different elements. From the update STEM-EDS mapping, the mapping images for Ir-L have obvious difference from that of Pt and Cu elements in the distribution through careful comparison. IrO_x looks like a random deposition on the entire surface of the SiC from Ir-L mapping. The distributions of Pt in the selected area are coincident and appear at almost the same position with Cu. This demonstrates that the deposited Pt is fully covered with Cu₂O at the same sites on SiC surface. Indeedly, besides the Cu₂O deposition on Pt, a part of Cu₂O is evenly distributed on SiC surface.*

Figure 2 (a) TEM, (b) HRTEM images and (c) STEM image and corresponding EDS mapping profiles for C-K, Si-K, Pt-L, Cu-K and Ir-L of $\text{Cu}_2\text{O-Pt/SiC/IrO}_x$.

Question 3. Some references on CO_2 photoreduction should be cited, such as Nature Communications 2018, 9, 1252 etc.

Response 3. Thanks for your suggestions, the relative references have cited in the revised manuscript.

Reviewers' Comments:

Reviewer #1:

Remarks to the Author:

The authors have carefully addressed my concerns with supplementary experimental characterizations and performance tests. The reasons for high selectivity to product HCOOH and the permeation phenomenon of redox shuttle $\text{Fe}^{2+}/\text{Fe}^{3+}$ are also proposed. I think the revised version can be accepted for publication in Nature Communication.

Reviewer #2:

Remarks to the Author:

The revised paper has been a great improvement in quality, and some minor revision still be needed before publishing in Nature Communications. My comments and suggestions are the following:

1. I think in this work there is still lack the strong evidence to support the statement "Cu₂O fully covered over Pt nanoparticles" and "core-shell structure" in the revised manuscript just from the present TEM images and EDS mapping results.
2. To be more clear, please add some explanations for the effect of amount of metals on the final photocatalytic results.
3. In the revised manuscript, Page 8 line 230, the authors claimed that the metals (Pt, Ir and Cu) keep almost the same before and after the reaction, however, according to the Figure S10, the XPS of metals show big difference between fresh and used state, especially for Pt a shoulder peak can be observed in the left peak. Thus, the XPS of these metals should be reanalyzed and point out these difference.
4. The Figure 2 and Figure 8 in this paper, brackets are missing.
5. Please keep the same Y axis titles of DRS spectra of the samples in this work.

Reviewer #3:

Remarks to the Author:

The revised manuscript can be acceptable for publication

Point-by-point Response to the reviewer 2#

Reviewer #2 (Remarks to the Author):

The revised paper has been a great improvement in quality, and some minor revision still be needed before publishing in Nature Communications. My comments and suggestions are the following:

Question 1. I think in this work there is still lack the strong evidence to support the statement “Cu₂O fully covered over Pt nanoparticles” and “core-shell structure” in the revised manuscript just from the present TEM images and EDS mapping results.

Response 1. *We are very grateful for your rigorous scholarship. Indeed, the TEM images and EDS mapping results are insufficient to support this statement. Taking into consideration of the difficulty of further analysis in both time and means, we changed the description “Cu₂O fully covered over Pt nanoparticles” into “Cu₂O was photocatalytically deposited over Pt nanoparticles to form an intimate contact configuration”, and the description “Cu₂O-Pt core-shell structure” into “Cu₂O-Pt intimate contact configuration”, in the revised manuscript. The deposition of Cu₂O on the Pt nanoparticle in Pt/SiC sample can be reasonable. This is because under UV light, the photogenerated electrons on SiC predominately transfer towards Pt where Cu²⁺ is reduced into Cu₂O. Moreover, the low energy ion scattering (LEIS) spectra (Figure 3a and 3b) of Pt/SiC and Cu₂O-Pt/SiC samples demonstrated that the Pt peak disappeared with the deposition of Cu₂O on Pt/SiC, which confirm the deposition of Cu₂O on Pt nanoparticle surface.*

Question 2. To be more clear, please add some explanations for the effect of amount of metals on the final photocatalytic results.

Response 2. *Thanks for your important suggestions. The final Cu₂O-Pt/SiC/IrO_x samples are prepared by dispersing Pt/SiC samples in aqueous solution containing both Cu²⁺ and IrCl₆³⁻ ions followed by light illumination for different time. Thus, the contents of Cu₂O and IrO_x on Pt/SiC surface are increased with increasing photodeposition time. With increasing the photodeposition*

time, the Cu_2O is gradually deposited onto Pt nanoparticle surface. The formed Cu_2O -Pt cocatalysts on SiC enhance the CO_2 reduction into HCOOH. The highest yield of HCOOH, $896.7 \mu\text{mol g}^{-1} \text{h}^{-1}$, over Cu_2O -Pt/SiC/ IrO_x occurs at ca. 1.8 wt% content of Cu_2O . The further increase of Cu_2O contents induces more amount of Cu_2O on SiC to block the optical absorption of SiC, and too thick Cu_2O layer on Pt surface is unfavorable to CO_2 reduction reaction on Cu_2O -Pt [Angew. Chem. Int. Ed. 2013, 52, 5776-5779]. With increasing the IrO_x contents, the more active sites of IrO_x are provided to enhance the Fe^{2+} oxidation. However, an excess amount of IrO_x may lead to the growth of IrO_x into large particles on SiC surface, and thus decrease the photocatalytic reaction. In the revised manuscript, we added the discussion “When Cu_2O content is higher than 1.8 wt%, further increasing Cu_2O photodeposition induces more amount of Cu_2O on both the Pt particles and the exposed SiC surface. But the deposited Cu_2O on the SiC could block the optical absorption of SiC, and a too thick Cu_2O layer on Pt surface is also unfavorable to CO_2 reduction reaction on Cu_2O -Pt³². As increasing IrO_x contents < 2.2 wt%, the more active sites of IrO_x are provided to enhance the Fe^{2+} oxidation. However, an excess amount of IrO_x may lead to the growth of IrO_x into large particles on SiC surface, and thus decrease the photocatalytic reaction.”

Question 3. In the revised manuscript, Page 8 line 230, the authors claimed that the metals (Pt, Ir and Cu) keep almost the same before and after the reaction, however, according to the Figure S10, the XPS of metals show big difference between fresh and used state, especially for Pt a shoulder peak can be observed in the left peak. Thus, the XPS of these metals should be reanalyzed and point out these difference.

Response 3. Thank you very much for your valuable comments, and very sorry for our previous rough conclusions. Indeed, the XPS spectra of metals show visible difference before and after the photocatalytic reaction, indicating some changes in the chemical states of sample during the photocatalytic process. However, the changes of metal states are likely to have little effect on the photocatalytic performance of sample based on the reasons as follows. (i) For Pt, the XPS Pt 4f peaks narrow down and the shoulder peaks become weak for the used sample, but the binding energy (BE) values of main peaks keep almost unchanged before and after the photocatalytic reaction. The main peaks at the BEs of 74.7 eV (Pt 4f 5/2) and 71.5 eV (Pt 4f 7/2) are attributed to

Pt(0). The shoulder peaks at 76.1 (or 77.2) eV (Pt 4f 5/2) and 72.9 (or 74.1) eV (Pt 4f 7/2) are belonged to Pt²⁺ (or Pt⁴⁺). It can be seen that most Pt atoms remain metallic states, and a part of high valence Pt species were translated also into Pt (0) after longtime reaction. The transformation to Pt(0) states after the photocatalytic reaction could be more beneficial to the photocatalysis. (ii) For IrO_x, the Ir 4f BE peaks not only become narrow but also shift towards lower energy after the photocatalytic reaction. Nevertheless, the wide Ir 4f peak of both the fresh and the used samples all cover the peaks of Ir⁰, Ir³⁺ and Ir⁴⁺ species, indicating the mix valence state feature of IrO_x. Accordingly, the Ir 4f_{7/2} peak for both the fresh and the used samples can be deconvoluted into a contribution of Ir⁰ (61.3 eV), Ir⁴⁺ (62.4 eV), and Ir³⁺ (63.5 eV) species [1-3]. The phenomena are universal for Ir-loaded catalysts, which is also the reason why iridium oxide is usually be expressed as IrO_x rather than IrO₂ [4]. It is estimated from the peak intensities that the contents of Ir⁴⁺ are increased, while Ir³⁺ contents are decreased in the used sample as compared with the fresh sample (seen Table S4). Such a change in the Ir 4f XPS spectra can be explained by the changes in the crystallinity and coordination numbers [5,6]. For the fresh sample, the broader and higher BE peaks suggest the existence of partial amorphous or high oxygen coordinated Ir species. For the used sample, the Ir 4f BE shift towards lower energy indicates an increase in the rutile phase IrO₂ during the photocatalytic process since the IrO_x could be mainly excited from the d(t2g) to the d(eg) band (1.5~2.75 eV) under visible light irradiation based on the literature [7]. It is possible that the change in Ir valence state or crystallinity do not affect the photoinduced d-d transition and thus photocatalytic performance. (iii) As for Cu₂O, the Cu 2p XPS spectrum shows a minor change. The BEs of the main peaks of Cu 2p remain almost the same before and after the reaction, only a minuscule shift towards lower energy. We could not exclude the possibility that a small amount of Cu(I) was translated into Cu(0) after the photocatalytic reaction. Because some Cu₂O is deposited synchronously on the SiC surface, the small change of Cu valence state can occur partly for these Cu₂O species, which could have no remarkable influence on the photocatalytic performance. These discussions have been added in the revised manuscript.

Figure S10. The XPS spectra of the Cu₂O-Pt/SiC/IrO_x sample before and after cycle reaction

Table S4. The contents of Ir³⁺, Ir⁴⁺ and Ir⁰ in Cu₂O-Pt/SiC/IrO_x sample before and after cycle reaction

	Ir ³⁺	Ir ⁴⁺	Ir ⁰
Content in used sample (%)	16.2	81.0	2.8
Content in fresh sample (%)	40.4	58.8	0.8

References

44. Cho, S. -H. et al. Synergistic Coupling of Metallic Cobalt Nitride Nanofibers and IrO_x Nanoparticle Catalysts for Stable Oxygen Evolution. *Chem. Mater.* **30**, 5941–5950 (2018).
45. Kwon, T. et al. Cobalt Assisted Synthesis of IrCu Hollow Octahedral Nanocages as Highly Active Electrocatalysts toward Oxygen Evolution Reaction. *Adv. Funct. Mater.* **27**, 1604688 (2017).
46. Pfeifer, V. et al. The electronic structure of iridium oxide electrodes active in water splitting. *Phys. Chem. Chem. Phys.* **18**, 2292–2296 (2016).
47. Li, P., Kong, L., Liu, J., Yan, J. & Liu, S. Photoassisted Hydrothermal Synthesis of IrO_x-TiO₂ for Enhanced Water Oxidation. *ACS Sustain. Chem. Eng.* **7**, 17941-17949 (2019).
48. Esquiús, J. R. et al. Effect of Base on the Facile Hydrothermal Preparation of Highly Active IrO_x Oxygen Evolution Catalysts. *ACS Appl. Energy Mater.* **3**, 800–809 (2020).
49. Cheng, H. et al. Aging amorphous/crystalline heterophase PdCu nanosheets for catalytic reactions. *Natl. Sci. Rev.* **6**, 955–961 (2019).
50. Frame, F. A. et al. Photocatalytic Water Oxidation with Nonsensitized IrO₂ Nanocrystals under Visible and UV Light. *J. Am. Chem. Soc.* **133**, 7264-7267 (2011).

Question 4. The Figure 2 and Figure 8 in this paper, brackets are missing.

Response 4. Thank a lot for mention. The brackets were added for the label of Figure 2 and 8.

Question 5. Please keep the same Y axis titles of DRS spectra of the samples in this work.

Response 5. *Thank you very much. The Y axis title of DRS spectra of Cu₂O samples was changed into “F(R)” the same as that of SiC samples (Figure S19)*

Reviewers' Comments:

Reviewer #2:

Remarks to the Author:

The authors have addressed all my concerns in the revised manuscript, and the paper can be accepted with present form.